# Implicit Bias of Gradient Descent on Reparametrized Models: On Equivalence to Mirror Descent

**Zhiyuan Li**[*]
Princeton University
Princeton, NJ 08540
zhiyuanli@cs.princeton.edu

**Tianhao Wang**[*]
Yale University
New Haven, CT 06511
tianhao.wang@yale.edu

**Jason D. Lee**
Princeton University
Princeton, NJ 08540
jasonlee@princeton.edu

**Sanjeev Arora**
Princeton University
Princeton, NJ 08540
arora@cs.princeton.edu

## Abstract

As part of the effort to understand implicit bias of gradient descent in over-parametrized models, several results have shown how the training trajectory on the overparametrized model can be understood as mirror descent on a different objective. The main result here is a characterization of this phenomenon under a notion termed *commuting parametrization*, which encompasses all the previous results in this setting. It is shown that gradient flow with any commuting parametrization is equivalent to continuous mirror descent with a related Legendre function. Conversely, continuous mirror descent with any Legendre function can be viewed as gradient flow with a related commuting parametrization. The latter result relies upon Nash's embedding theorem.

## 1 Introduction

*Implicit bias* refers to the phenomenon in machine learning whereby the solution obtained from loss minimization has special properties that were not implied by value of the loss function and instead arose from the optimization's trajectory through the parameter space. Quantifying implicit bias necessarily has to go beyond the traditional black-box convergence analyses of optimization algorithms. Implicit bias can explain how choice of optimization algorithm can affect generalization [61, 42, 41].

Many existing results about implicit bias view training (in the limit of infinitesimal step size) as a differential equation or process $\{x(t)\}_{t\geq 0} \subset \mathbb{R}^D$. To show the implicit bias of $x(t)$, the idea is to show for another (more intuitive or better understood) process $\{w(t)\}_{t\geq 0} \subset \mathbb{R}^d$ that $x(t)$ is *simulating* $w(t)$, in the sense that there exists a mapping $G : \mathbb{R}^D \to \mathbb{R}^d$ such that $w(t) = G(x(t))$. Then the implicit bias of $x(t)$ can be characterized by translating the special properties of $w(t)$ back to $x(t)$ through $G$. A related term, *implicit regularization*, refers to a handful of such results where particular update rules are shown to lead to regularized solutions; specifically, $x(t)$ is simulating $w(t)$ where $w(t)$ is solution to a regularized version of the original loss.

The current paper develops a general framework involving optimization in the continuous-time regime of a loss $L : \mathbb{R}^d \to \mathbb{R}$ that has been re-parametrized before optimization as $w = G(x)$ for some $G : \mathbb{R}^D \to \mathbb{R}^d$. Then the original loss $L(w)$ in the $w$-space induces the implied loss

---

[*]Equal contribution

36th Conference on Neural Information Processing Systems (NeurIPS 2022).

$(L \circ G)(x) \equiv L(G(x))$ in the $x$-space, and the gradient flow in the $x$-space is given by[2]

$$\mathrm{d}x(t) = -\nabla(L \circ G)(x(t))\mathrm{d}t. \qquad (1)$$

Using $w(t) = G(x(t))$ and the fact that $\nabla(L \circ G)(x) = \partial G(x)^\top \nabla L(G(x))$ where $\partial G(x) \in \mathbb{R}^{d \times D}$ denotes the Jacobian of $G$ at $x$, the corresponding dynamics of (1) in the $w$-space is

$$\mathrm{d}w(t) = \partial G(x(t))\mathrm{d}x(t) = -\partial G(x(t))\partial G(x(t))^\top \nabla L(w(t))\mathrm{d}t. \qquad (2)$$

Our framework is developed to fully understand phenomena in recent papers [26, 58, 64, 4, 61, 5, 8], which give examples suggesting that gradient flow in the $x$-space could end up simulating a more classical algorithm, mirror descent (specifically, the continuous analog, mirror flow) in the $w$-space. Recall that mirror flow is continuous-time limit of the classical mirror descent, written as $\mathrm{d}\nabla R(w(t)) = -\nabla L(w(t))\mathrm{d}t$ where $R : \mathbb{R}^d \to \mathbb{R} \cup \{\infty\}$ is a strictly convex function [49, 10], which is called *mirror map* or *Legendre function* in literature. Equivalently it is *Riemannian gradient flow* with metric tensor $\nabla^2 R$, an old notion in geometry:

$$\mathrm{d}w(t) = -\nabla^2 R(w(t))^{-1} \nabla L(w(t))\mathrm{d}t. \qquad (3)$$

If there exists a Legendre function $R$ such that $\partial G(x(t))\partial G(x(t))^\top = \nabla^2 R(w(t))^{-1}$ for all $t$, then (2) becomes a simple mirror flow in the $w$-space. Many existing results about implicit bias indeed concern reparametrizations $G$ that satisfy $\partial G(x)\partial G(x)^\top = \nabla^2 R(w)^{-1}$ for a strictly convex function $R$, and the implicit bias/regularization is demonstrated by showing that the convergence point satisfies the KKT conditions needed for minimizing $R$ among all minimizers of the loss $L$. A concrete example is that $w_i(t) = G_i(x(t)) = (x_i(t))^2$ for all $i \in [d]$, so here $D = d$. In this case, the Legendre function $R$ must satisfy $(\nabla^2 R(w(t)))^{-1} = \partial G(x(t))\partial G(x(t))^\top = 4\mathrm{diag}((x_1(t))^2, \ldots, (x_d(t))^2) = 4\mathrm{diag}(w_1(t), \ldots, w_d(t))$ which suggests $R$ is the classical negative entropy function, *i.e.*, $R(w) = \sum_{i=1}^d w_i(\ln w_i - 1)$.

However, in general, it is hard to decide *whether gradient flow for a given parametrization $G$ can be written as mirror flow for some Legendre function $R$*, especially when $D > d$ and $G$ is not an injective map. In such cases, there could be multiple $x$'s mapping to the same $G(x)$ yet having different $\partial G(x)\partial G(x)^\top$. If more than one of such $x$ can be reached by gradient flow, then the desired Legendre function cannot exist.[3] If only one of such $x$ can be reached by gradient flow, we must decide which $x$ it is in order to decide the value of $\nabla^2 R$ using $\partial G \partial G^\top$. Conversely, [5] raises the following question: *for what Legendre function $R$ can the corresponding mirror flow be the result of gradient flow after some reparametrization $G$?* Answering the questions in both directions requires a deeper understanding of the impact of parametrizations.

The following are the main contributions of the current paper:

- In Section 4, building on classic study of commuting vector fields we identify a notion of when a parametrization $w = G(x)$ is *commuting* (Definition 4.1) and use it to give a sufficient condition (Theorem 4.8) and a slightly weaker necessary condition (Theorem 4.9) of when the gradient flow in the $x$-space governed by $-\nabla(L \circ G)$ is simulating a mirror flow in the $w$-space with respect to some Legendre function $R : \mathbb{R}^d \to \mathbb{R}$. This condition encompasses all the previous results [26, 58, 64, 4, 61, 5, 8]. Moreover, the Legendre function is independent of the loss $L$ and depends only on the initialization $x_{\mathrm{init}}$ and the parametrization $G$.

- We recover and generalize existing implicit bias results for underdetermined linear regression as implications of the above characterization (Corollary 4.17). We also give new convergence analysis in such settings (Theorem 4.15), filling the gap in previous works [26, 61, 8] where parameter convergence is only assumed but not proved.

- In the reverse direction, we use the famous Nash's embedding theorem to show that every mirror flow in the $w$-space with respect to some Legendre function $R$ simulates a gradient flow with commuting parametrization under some embedding $x = F(w)$ where $F : \mathbb{R}^d \to \mathbb{R}^D$ and the

---

[2]Two examples from recent years, where $G$ does not change expressiveness of the model, involve (a) overparametrized linear regression where the parameter vector $w$ is reparametrized (for example as $w = u^{\odot 2} - v^{\odot 2}$ [61]) and (b) deep linear nets [6] where a matrix $W$ is factorized as $W = W_1 W_2 \cdots W_L$ where each $W_\ell$ is the weight matrix for the $\ell$-th layer.

[3]To avoid such an issue, [5] has to assume all the preimages of $G$ at $w$ have the same $\partial G(\partial G)^\top$ and a recent paper [23] assumes that $G$ is injective.

parametrization $G$ is the inverse of $F$ (Theorem 5.1). This provides an affirmative and fully general answer to the question of when such reparametrization functions exist, giving a full answer to questions raised in a more restricted setting in [5].

## 2  Related work

**Implicit bias.** With high overparametrization as used in modern machine learning, there usually exist multiple optima, and it is crucial to understand which particular solutions are found by the optimization algorithm. Implicit bias of gradient descent for classification tasks with separable data was studied in [55, 24, 46, 35, 45, 34] and for non-separable data in [32, 33], where the implicit bias appears in the form of margin maximization. The implicit bias for regression problems has also been analyzed by leveraging tools like mirror descent [61, 24, 64, 58, 4, 5], later generalized in [8].

The sharp contrast between the so-called *kernel* and *rich* regimes [61] reflects the importance of the initialization scale, where a large initialization often leads to the kernel regime with features barely changing during training [30, 16, 20, 19, 2, 1, 65, 7, 62, 31], while with a small initialization, the solution exhibits richer behavior with the resulting model having lower complexity [25, 26, 39, 52, 6, 15, 41, 43, 44, 53, 56, 22]. Recently [63] gave a complete characterization on the relationship between initialization scale, parametrization and learning rate in order to avoid kernel regime.

There are also papers on the implicit bias of other types of optimization algorithms, e.g., stochastic gradient descent [40, 11, 29, 42, 18, 66] and adaptive and momentum methods [51, 60, 59, 36], to name a few.

**Understanding mirror descent.** In the continuous-time regime, the mirror flow is equivalent to a Riemannian gradient flow with the metric tensor induced by the Legendre function. [27] showed that a partial discretization of the latter gives rise to the classical mirror descent. Assuming the existence of some reparametrization function, [5] showed that a particular mirror flow can be reparametrized as a gradient flow. Our paper shows that such reparametrization always exists by using Nash's embedding theorem. [23] generalized the equivalence result of [5] to discrete updates.

## 3  Preliminaries and notations

**Notations.** We denote $\mathbb{N}$ as the set of natural numbers. For any $n \in \mathbb{N}$, we denote $\{1, 2, \ldots, n\}$ by $[n]$. For any vector $u \in \mathbb{R}^D$, we denote its $i$-th coordinate by $u_i$. For any vector $u, v \in \mathbb{R}^D$ and $\alpha \in \mathbb{R}$, we define $u \odot v = (u_1 v_1, \ldots, u_D v_D)^\top$ and $u^{\odot \alpha} = ((u_1)^\alpha, \ldots, (u_D)^\alpha)^\top$. For any $k \in \mathbb{N} \cup \{\infty\}$, we say a function $f$ is $\mathcal{C}^k$ if it is $k$ times continuously differentiable, and use $\mathcal{C}^k(M)$ to denote the set of all $C^k$ functions from $M$ to $\mathbb{R}$. We use $\circ$ to denote the composition of functions, *e.g.*, $f \circ g(x) = f(g(x))$. For any convex function $R : \mathbb{R}^D \to \mathbb{R} \cup \{\infty\}$, we denote its domain by $\mathrm{dom}\, R = \{w \in \mathbb{R}^D | R(w) < \infty\}$. For any set $S$, we denote its interior by $\mathrm{int}(S)$ and its closure by $\overline{S}$.

We assume that the model has parameter vector $w \in \mathbb{R}^d$ and $\mathcal{C}^1$ loss function $L : \mathbb{R}^d \to \mathbb{R}$. Training involves a reparametrized vector $x \in \mathbb{R}^D$, which is a reparametrization of $w$ such that $w = G(x)$ for some differentiable parametrization function $G$, and the objective is $L(G(x))$. From now on, we follow the convention that $d$ is the dimension of the original parameter $w$ and $D$ is the dimension of the reparametrized $x$. We also refer to $\mathbb{R}^d$ as the $w$-space and $\mathbb{R}^D$ as the $x$-space.

In particular, we are interested in understanding the dynamics of gradient flow under the objective $L \circ G$ on some submanifold $M \subseteq \mathbb{R}^D$. Most of our results also generalize to the following notion of *time-dependent* loss.

**Definition 3.1** (Time-dependent loss)**.** A time-dependent loss $L_t(w)$ is a function piecewise constant in time $t$ and continuously differentiable in $w \in \mathbb{R}^d$, that is, there exist $k \in \mathbb{N}$, $0 = t_1 < t_2 < \cdots < t_{k+1} = \infty$ and $\mathcal{C}^1$ loss functions $L^{(1)}, L^{(2)}, \ldots, L^{(k)}$ such that for each $i \in [k]$ and all $t \in [t_i, t_{i+1})$,

$$L_t(w) = L^{(i)}(w), \qquad \forall w \in \mathbb{R}^d.$$

We denote the set of such time-dependent loss functions by $\mathcal{L}$.

## 3.1 Manifold and vector field

Vector fields are a natural way to formalize the continuous-time gradient descent (a good reference is [38]). Let $M$ be any smooth submanifold of $\mathbb{R}^D$. A *vector field* $X$ on $M$ is a continuous map from $M$ to $\mathbb{R}^D$ such that for any $x \in M$, $X(x)$ is in the tangent space of $M$ at $x$, which is denoted by $T_x M$. Formally, $T_x M := \left\{ \frac{\mathrm{d}\gamma}{\mathrm{d}t}\big|_{t=0} \mid \forall \text{ smooth curves } \gamma : \mathbb{R} \to M, \gamma(0) = x \right\}$.

**Definition 3.2** (Complete vector field; p.215, [38]). Let $M$ be a smooth submanifold of $\mathbb{R}^D$ and $X$ be a vector field on $M$. We say $X$ is a *complete vector field* on $M$ if for any initialization $x_{\mathrm{init}} \in M$, the differential equation $\mathrm{d}x(t) = X(x(t))\mathrm{d}t$ has a solution on $(-\infty, \infty)$ with $x(0) = x_{\mathrm{init}}$.

Equipping the smooth submanifold $M \subseteq \mathbb{R}^D$ with a metric tensor $g$, we then have a Riemannian manifold $(M, g)$, where for each $x \in M$, $g_x : T_x M \times T_x M \to \mathbb{R}$ is a positive definite bilinear form. In particular, the standard Euclidean metric $\overline{g}$ corresponds to $\overline{g}_x(u, v) = u^\top v$ for each $x \in M$ and $u, v \in T_x M$, under which the length of any arc on $M$ is given by its length as a curve in $\mathbb{R}^D$.

For any differentiable function $f : M \to \mathbb{R}$, we denote by $\nabla_g f$ its gradient vector field with respect to metric tensor $g$. More specifically, $\nabla_g f(x)$ is defined as the unique vector in $\mathbb{R}^D$ such that $\nabla_g f(x) \in T_x M$ and $\frac{\mathrm{d}f(\gamma(t))}{\mathrm{d}t}\big|_{t=0} = g_x\big(\nabla f(x), \frac{\mathrm{d}\gamma(t)}{\mathrm{d}t}\big|_{t=0}\big)$ for any smooth curve $\gamma : \mathbb{R} \to M$ with $\gamma(0) = x$. Throughout the paper, we assume by default that the metric on the submanifold $M \subseteq \mathbb{R}^D$ is inherited from $(\mathbb{R}^D, \overline{g})$, and we will use $\nabla f$ as a shorthand for $\nabla_{\overline{g}} f$. If $M$ is an open set of $\mathbb{R}^D$, $\nabla f$ is then simply the ordinary gradient of $f$.

For any $x \in M$ and $\mathcal{C}^1$ function $f : M \to \mathbb{R}$, we denote by $\phi_f^t(x)$ the point on $M$ reached after time $t$ by following the vector field $-\nabla f$ starting at $x$, *i.e.*, the solution at time $t$ (when it exists) of

$$\mathrm{d}\phi_f^t = -\nabla f(\phi_f^t)\mathrm{d}t, \qquad \phi_f^0(x) = x.$$

We say $\phi_f^t(x)$ is *well-defined* at time $t$ when the above differential equation has a solution at time $t$. Moreover, for any differentiable function $X : M \to \mathbb{R}^d$, we define its Jacobian by

$$\partial X(x) = (\nabla X_1(x), \nabla X_2(x), \dots, \nabla X_d(x))^\top.$$

**Definition 3.3** (Lie bracket). Let $M$ be a smooth submanifold of $\mathbb{R}^D$. Given two $C^1$ vector fields $X, Y$ on $M$, we define the *Lie bracket* of $X$ and $Y$ as $[X, Y](x) := \partial Y(x) X(x) - \partial X(x) Y(x)$.

## 3.2 Parametrizations

We use the term *parametrization* to refer to differentiable maps from a smooth submanifold of $\mathbb{R}^D$ ($x$-space) to $\mathbb{R}^d$ ($w$-space). We reserve $G$ to denote parametrizations, and omit the dependence on $G$ for notations of objects related to $G$ when it is clear from the context.

The following notion of regular parametrization plays an important role in our analysis, and it is necessary for our main equivalence result between mirror flow and gradient flow with commuting parametrization. This is because if the null space of $\partial G(x)$ is non-trivial, *i.e.,* it contains some vector $u \neq 0$, then the gradient flow with parametrization $G$ obviously cannot simulate any mirror flow with nonzero velocity in the direction of $u$.

**Definition 3.4** (Regular parametrization). Let $M$ be a smooth submanifold of $\mathbb{R}^D$. A *regular parametrization* $G : M \to \mathbb{R}^d$ is a $\mathcal{C}^1$ parametrization such that $\partial G(x)$ is of rank $d$ for all $x \in M$.

Note that a regular parametrization $G$ can become irregular when its domain is changed. For example, $G(x) = x^2$ is regular on $\mathbb{R}_+$, but it is not regular on $\mathbb{R}$ as $\partial G(0) = 0$.

Given a $\mathcal{C}^2$ parametrization $G : M \to \mathbb{R}^d$, for any $x \in M$ and $\mu \in \mathbb{R}^d$, we define

$$\psi(x; \mu) := \phi_{G_1}^{\mu_1} \circ \phi_{G_2}^{\mu_2} \circ \cdots \circ \phi_{G_d}^{\mu_d}(x) \tag{4}$$

when it is well-defined, *i.e.*, the corresponding integral equation has a solution. For any $x \in M$, we define the domain of $\psi(x; \cdot)$ as

$$\mathcal{U}(x) = \big\{ \mu \in \mathbb{R}^d \mid \psi(x; \mu) \text{ is well-defined} \big\}. \tag{5}$$

When every $\nabla G_i$ is a complete vector field on $M$ as in Definition 3.2, we have $\mathcal{U}(x) = \mathbb{R}^d$. However, such completeness assumption is relatively strong, and most polynomials would violate it. For

example, consider $G(x) = x^{\odot 3}$ for $x \in \mathbb{R}^d$, then the solution to $\mathrm{d}x_i(t) = 3x_i(t)^2\mathrm{d}t$ explodes in finite time for each $i \in [d]$. To relax this, we consider parametrizations such that the domain of the flows induced by its gradient vector fields is pairwise symmetric. More specifically, we define

$$\mathcal{U}_{ij}(x) = \left\{(s,t) \in \mathbb{R}^2 \mid \phi_{G_i}^s \circ \phi_{G_j}^t(x) \text{ is well-defined}\right\}$$

for any $x \in M$ and $i,j \in [d]$, and we make the following assumption.

**Assumption 3.5.** Let $M$ be a smooth submanifold of $\mathbb{R}^D$ and $G : M \to \mathbb{R}^d$ be a parametrization. We assume that for any $x \in M$ and $i \in [d]$, $\phi_x^t(x)$ is well-defined for $t \in (T_-, T_+)$ such that either $\lim_{t \to T_+} \|\phi_x^t(x)\|_2 = \infty$ or $T_+ = \infty$ and similarly for $T_-$. Also, we assume that for any $x \in M$ and $i,j \in [d]$, it holds that $\mathcal{U}_{ji}(x) = \{(t,s) \in \mathbb{R}^2 \mid (s,t) \in \mathcal{U}_{ij}(x)\}$, i.e., $\phi_{G_i}^s \circ \phi_{G_j}^t(x)$ is well-defined if and only if $\phi_{G_j}^t \circ \phi_{G_i}^s(x)$ is.

Indeed, under Assumption 3.5, we can show that for any $x \in M$, $\mathcal{U}(x)$ is a hyperrectangle in $\mathbb{R}^d$, i.e,

$$\mathcal{U}(x) = \mathcal{I}_1(x) \times \mathcal{I}_2(x) \times \cdots \times \mathcal{I}_d(x) \quad \text{where each } \mathcal{I}_j(x) \subset \mathbb{R} \text{ is an open interval.} \tag{6}$$

See Lemma C.1 and its proof in Appendix C. Next, for any initialization $x_{\text{init}} \in M$, the set of points that are reachable via gradient flow under some time-dependent loss (see Definition 3.1) with parametrization $G$ is a subset of $M$ that depends on $G$ and $x_{\text{init}}$.

**Definition 3.6** (Reachable set). Let $M$ be a smooth submanifold of $\mathbb{R}^D$. For any $\mathcal{C}^2$ parametrization $G : M \to \mathbb{R}^d$ and any initialization $x_{\text{init}} \in M$, the reachable set $\Omega_x(x_{\text{init}}; G)$ is defined as

$$\Omega_x(x_{\text{init}}; G) = \left\{\phi_{L_1 \circ G}^{\mu_1} \circ \phi_{L_2 \circ G}^{\mu_2} \circ \cdots \circ \phi_{L_k \circ G}^{\mu_k}(x_{\text{init}}) \,\Big|\, \forall k \in \mathbb{N}, \forall i \in [k], L_i \in \mathcal{C}^1(\mathbb{R}^d), \mu_i \geq 0\right\}.$$

It is clear that the above definition induces a transitive "reachable" relationship between points on $M$, and it is also reflexive since for all $L \in \mathcal{C}^1(\mathbb{R}^d)$ and $t > 0$, $\phi_{L \circ G}^t \circ \phi_{(-L) \circ G}^t$ is the identity map on the domain of $\phi_{-L \circ G}^t$. In this sense, the reachable sets are orbits of the family of gradient vector fields $\{\nabla(L \circ G) \mid L \in \mathcal{C}^1(\mathbb{R}^d)\}$, *i.e.*, the reachable sets divide the domain $M$ into equivalent classes. The above reachable set in the $x$-space further induces the corresponding reachable set in the $w$-space given by $\Omega_w(x_{\text{init}}; G) = G(\Omega_x(x_{\text{init}}; G))$.

In most natural examples, the parametrization $G$ is smooth (though this is not necessary for our results), and by Sussman's Orbit Theorem [57], each reachable set $\Omega_x(x_{\text{init}}; G)$ is an immersed submanifold of $M$. Moreover, it follows that $\Omega_x(x_{\text{init}}; G)$ can be generated by $\{\nabla G_i\}_{i=1}^d$, *i.e.*, $\Omega_x(x_{\text{init}}; G) = \{\phi_{G_{j_1}}^{\mu_1} \circ \phi_{G_{j_2}}^{\mu_2} \circ \cdots \circ \phi_{G_{j_k}}^{\mu_k}(x_{\text{init}}) \mid \forall k \in \mathbb{N}, \forall i \in [k], j_i \in [d], \mu_i \geq 0\}$.

### 3.3 Mirror descent and mirror flow

Next, we introduce some basic notions for mirror descent [49, 10]. We refer the readers to Appendix B for more preliminaries on convex analysis.

**Definition 3.7** (Legendre function and mirror map). Let $R : \mathbb{R}^d \to \mathbb{R} \cup \{\infty\}$ be a differentiable convex function. We say $R$ is a *Legendre function* when it satisfies that (1) $R$ is strictly convex on $\text{int}(\text{dom}\,R)$, and (2) for any sequence $\{w_i\}_{i=1}^\infty$ going to the boundary of $\text{dom}\,R$, $\lim_{i \to \infty} \|\nabla R(w_i)\|_2 = \infty$. In particular, we call $R$ a *mirror map* if $R$ further satisfies that the gradient map $\nabla R : \text{int}(\text{dom}\,R) \to \mathbb{R}^d$ is surjective (see p.298 in [13]).

Given a Legendre function $R : \mathbb{R}^d \to \mathbb{R} \cup \{\infty\}$, for any initialization $w_0 = w_{\text{init}} \in \text{int}(\text{dom}\,R)$, mirror descent with step size $\eta$ updates as follows:

$$\nabla R(w_{k+1}) = \nabla R(w_k) - \eta \nabla L(w_k). \tag{7}$$

Usually $\nabla R$ is required to be surjective so that after a discrete descent step in the dual space, it can be projected back to the primal space via $(\nabla R)^{-1}$. Nonetheless, as long as $\nabla R(w_k) - \eta \nabla L(w_k)$ is in the range of $\nabla R$, the above discrete update is well-defined. In the limit of $\eta \to 0$, (7) becomes the continuous mirror flow:

$$\mathrm{d}\nabla R(w(t)) = -\nabla L(w(t))\mathrm{d}t. \tag{8}$$

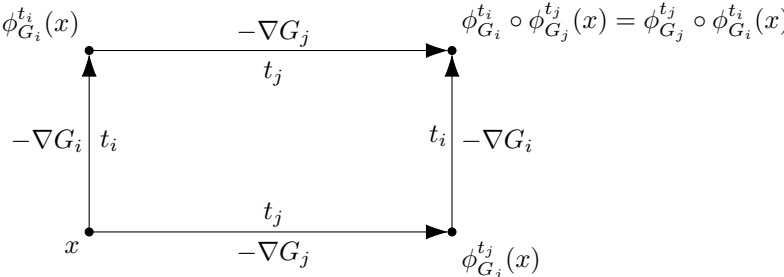

Figure 1: Illustration of commuting parametrizations. Suppose $G : M \to \mathbb{R}^d$ is a commuting parametrization satisfying Assumption 3.5, then starting from any $x \in M$, first moving along $-\nabla G_i$ for time $t_i$ then moving along $-\nabla G_j$ for time $t_j$ yields the same result as first moving along $-\nabla G_j$ for time $t_j$ then moving along $-\nabla G_i$ for time $t_i$ does, i.e., $\phi_{G_i}^{t_i} \circ \phi_{G_j}^{t_j}(x) = \phi_{G_j}^{t_j} \circ \phi_{G_i}^{t_i}(x)$.

Given a differentiable function $R$, the corresponding Bregman divergence $D_R$ is defined as

$$D_R(w, w') = R(w) - R(w') - \langle \nabla R(w'), w - w' \rangle.$$

We recall a well-known implicit bias result for mirror flow [24] (which holds for mirror descent as well), which shows that for a specific type of loss, if mirror flow converges to some optimal solution, then the convergence point minimizes some convex regularizer among all optimal solutions.

**Theorem 3.8.** *Given any data $Z \in \mathbb{R}^{n \times d}$ and corresponding label $Y \in \mathbb{R}^n$, suppose the loss $L(w)$ is in the form of $L(w) = \widetilde{L}(Zw)$ for some differentiable $\widetilde{L} : \mathbb{R}^n \to \mathbb{R}$. Assume that initialized at $w(0) = w_{\text{init}}$, the mirror flow (8) converges and the convergence point $w_\infty = \lim_{t \to \infty} w(t)$ satisfies $Zw_\infty = Y$, then $D_R(w_\infty, w_0) = \min_{w : Zw = Y} D_R(w, w_0)$.*

See Appendix C for a proof. The above theorem is the building block for proving the implicit bias induced by any commuting parametrization in overparametrized linear models (see Theorem 4.16).

## 4 Every gradient flow with commuting parametrization is a mirror flow

### 4.1 Commuting parametrization

We now formalize the notion of commuting parametrization. We remark that $M$ is a smooth submanifold of $\mathbb{R}^D$, and it is the domain of the parametrization $G$.

**Definition 4.1** (Commuting parametrization)**.** Let $M$ be a smooth submanifold of $\mathbb{R}^D$. A $\mathcal{C}^2$ parametrization $G : M \to \mathbb{R}^d$ is *commuting* in a subset $S \subseteq M$ if and only if for any $i, j \in [d]$, the Lie bracket $[\nabla G_i, \nabla G_j](x) = 0$ for all $x \in S$. Moreover, we say $G$ is a *commuting parametrization* if it is commuting in the entire $M$.

In particular, when $M$ is an open subset of $\mathbb{R}^d$, $\{\nabla G_i\}_{i=1}^d$ are ordinary gradients in $\mathbb{R}^D$, and the Lie bracket between any pair of $\nabla G_i$ and $\nabla G_j$ is given by $[\nabla G_i, \nabla G_j](x) = \nabla^2 G_j(x) \nabla G_i(x) - \nabla^2 G_i(x) \nabla G_j(x)$. This provides an easy way to check whether $G$ is commuting or not.

The above definition of commuting parametrizations builds upon the differential properties of the gradient vector fields $\{\nabla G_i\}_{i=1}^d$, where each Lie bracket $[\nabla G_i, \nabla G_j]$ quantifies the change of $\nabla G_j$ along the flow generated by $\nabla G_i$. Indeed, the above characterization of 'commuting' is further equivalent to another characterization in the integral form (Theorem 4.2), as illustrated in Figure 1.

**Theorem 4.2.** *Let $M$ be a smooth submanifold of $\mathbb{R}^D$ and $G : M \to \mathbb{R}^d$ be a $\mathcal{C}^2$ parametrization satisfying Assumption 3.5. For any $i, j \in [d]$, $[\nabla G_i, \nabla G_j](x) = 0$ for all $x \in M$ if and only if for any $x \in M$, it holds that $\phi_{G_i}^s \circ \phi_{G_j}^t(x) = \phi_{G_j}^t \circ \phi_{G_i}^s(x)$ for all $(s, t) \in \mathcal{I}_i(x) \times \mathcal{I}_j(x)$, where $\mathcal{I}_i(x)$ and $\mathcal{I}_j(x)$ are the time domains of $\phi_{G_i}^s(x)$ and $\phi_{G_j}^t(x)$ as defined in (6).*

The commuting condition clearly holds when each $G_i$ only depends on a different subset of coordinates of $x$, because we then have $\nabla^2 G_i(\cdot) \nabla G_j(\cdot) \equiv 0$ for any distinct $i, j \in [d]$ as $\nabla^2 G_i$ and $\nabla G_j$ live in different subspaces of $\mathbb{R}^D$. We call such $G$ *separable parametrizations*[4], and this case covers all the previous examples [26, 58, 4, 61, 5]. Another interesting example is the *quadratic parametrization*: We parametrize $w \in \mathbb{R}^d$ by $G : \mathbb{R}^D \to \mathbb{R}^d$ where for each $i \in [d]$, there is a symmetric matrix

---

[4]We further discuss the existence of non-separable commuting parametrizations in Appendix A.2.

$A_i \in \mathbb{R}^{D \times D}$ such that $G_i(x) = \frac{1}{2}x^\top A_i x$. Then each $[\nabla G_i, \nabla G_j](x) = (A_j A_i - A_i A_j)x$, and thus $G$ is a commuting parametrization if and only if matrices $\{A_i\}_{i=1}^d$ commute.

For concreteness, we analyze two examples below. The first one is both a separable parametrization and a commuting quadratic parametrization, while the second one is quadratic but non-commuting.

**Example 4.3** ($u^{\odot 2} - v^{\odot 2}$ parametrization, [61]). Parametrize $w \in \mathbb{R}^d$ by $w = u^{\odot 2} - v^{\odot 2}$. Here $D = 2d$, and the parametrization $G$ is given by $G(x) = u^{\odot 2} - v^{\odot 2}$ for $x = \binom{u}{v} \in \mathbb{R}^D$. Since each $G_i(x)$ involves only $u_i$ and $v_i$, $G$ is a separable parametrization and hence a commuting parametrization. Meanwhile, each $G_i(x)$ is a quadratic form in $x$, and it can be directly verified that the matrices underlying these quadratic forms commute with each other.

**Example 4.4** (Matrix factorization). As a counter-example, consider two parametrizations for matrix factorization: $G(U) = UU^\top$ and $G(U, V) = UV^\top$, where $U, V \in \mathbb{R}^{d \times r}$ and $d \geq 2, r \geq 1$. These are both *non-commuting* quadratic parametrizations. Here we only demonstrate for the parametrization $G(U) = UU^\top$, and $G(U, V) = UV^\top$ follows a similar argument. For each $i, j \in [d]$, we define $E_{ij} \in \mathbb{R}^d$ as the one-hot matrix with the $(i, j)$-th entry being 1 and the rest being 0, and denote $\overline{E}_{ij} = \frac{1}{2}(E_{ij} + E_{ji})$. For $r = 1$, we have $G_{ij}(U) = U_i U_j = U^\top \overline{E}_{ij} U$ for any $i, j \in [d]$, so $G$ is a quadratic parametrization. Note that $\overline{E}_{ii}\overline{E}_{ij} = \frac{1}{2}E_{ij} \neq \frac{1}{2}E_{ji} = \overline{E}_{ij}\overline{E}_{ii}$ for all distinct $i, j \in [d]$, which implies that $[\nabla G_{ij}, \nabla G_{ii}] \neq 0$, so $G$ is non-commuting. More generally, we can reshape $U$ as a vector $\overrightarrow{U} := [U_{:1}^\top, \ldots, U_{:r}^\top]^\top \in \mathbb{R}^{rd}$ where each $U_{:j}$ is the $j$-th column of $U$, and the resulting quadratic form for the $(i, j)$-entry of $G(U)$ corresponds to a block-diagonal matrix:

$$G_{ij}(U) = (\overrightarrow{U})^\top \mathrm{diag}(\overline{E}_{ij}, \ldots, \overline{E}_{ij})\overrightarrow{U}.$$

Therefore, $\nabla^2 G_{ij}$ does not commute with $\nabla^2 G_{ii}$ due to the same reason as in the rank-1 case.

**Remark 4.5.** *This non-commuting issue for general matrix factorization does not conflict with the theoretical analysis in [26] where the measurements are commuting, or equivalently, only involve diagonal elements, as $\{G_{ii}\}_{i=1}^d$ are indeed commuting parametrizations. [26] is the first to identify the above non-commuting issue and conjectured that the implicit bias result for diagonal measurements can be extended to the general case.*

### 4.2 Main equivalence result

Next, we proceed to present our analysis for gradient flow with commuting parametrization. The following two lemmas highlight the special properties of commuting parametrizations. Lemma 4.6 shows that the point reached by gradient flow with any commuting parametrization is determined by the integral of the negative gradient of the loss along the trajectory.

**Lemma 4.6.** *Let $M$ be a smooth submanifold of $\mathbb{R}^D$ and $G : M \to \mathbb{R}^d$ be a commuting parametrization. For any initialization $x_{\mathrm{init}} \in M$, consider the gradient flow for any time-dependent loss $L_t \in \mathcal{L}$ as in Definition 3.1: $\mathrm{d}x(t) = -\nabla(L_t \circ G)(x(t))\mathrm{d}t$, $x(0) = x_{\mathrm{init}}$. Further define $\mu(t) = \int_0^t -\nabla L_t(G(x(s)))\mathrm{d}s$. Suppose $\mu(t) \in \mathcal{U}(x_{\mathrm{init}})$ for all $t \in [0, T)$ where $T \in \mathbb{R} \cup \{\infty\}$, then it holds that $x(t) = \psi(x_{\mathrm{init}}; \mu(t))$ for all $t \in [0, T)$.*

Based on Lemma 4.6, the next key lemma reveals the essential approach to find the Legendre function.

**Lemma 4.7.** *Let $M$ be a smooth submanifold of $\mathbb{R}^D$ and $G : M \to \mathbb{R}^d$ be a commuting and regular parametrization satisfying Assumption 3.5. Then for any $x_{\mathrm{init}} \in M$, there exists a Legendre function $Q : \mathbb{R}^d \to \mathbb{R} \cup \{\infty\}$ such that $\nabla Q(\mu) = G(\psi(x_{\mathrm{init}}; \mu))$ for all $\mu \in \mathcal{U}(x_{\mathrm{init}})$. Moreover, let $R$ be the convex conjugate of $Q$, then $R$ is also a Legendre function and $\mathrm{int}(\mathrm{dom}\,R) = \Omega_w(x_{\mathrm{init}}; G)$ and $\nabla^2 R(G(\psi(x_{\mathrm{init}}; \mu))) = (\partial G(\psi(x_{\mathrm{init}}; \mu))\partial G(\psi(x_{\mathrm{init}}; \mu))^\top)^{-1}$ for all $\mu \in \mathcal{U}(x_{\mathrm{init}})$.*

Next, we present our main result on characterization of gradient flow with commuting parametrization.

**Theorem 4.8.** *Let $M$ be a smooth submanifold of $\mathbb{R}^D$ and $G : M \to \mathbb{R}^d$ be a commuting and regular parametrization satisfying Assumption 3.5. For any initialization $x_{\mathrm{init}} \in M$, consider the gradient flow for any time-dependent loss function $L_t : \mathbb{R}^d \to \mathbb{R}$:*

$$\mathrm{d}x(t) = -\nabla(L_t \circ G)(x(t))\mathrm{d}t, \qquad x(0) = x_{\mathrm{init}}.$$

*Define $w(t) = G(x(t))$ for all $t \geq 0$, then the dynamics of $w(t)$ is a mirror flow with respect to the Legendre function $R$ given by Lemma 4.7, i.e.,*

$$\mathrm{d}\nabla R(w(t)) = -\nabla L_t(w(t))\mathrm{d}t, \qquad w(0) = G(x_{\mathrm{init}}).$$

*Moreover, this $R$ only depends on the initialization $x_{\text{init}}$ and the parametrization $G$, and is independent of the loss function $L_t$.*

Theorem 4.8 provides a sufficient condition for when a gradient flow with certain parametrization $G$ is simulating a mirror flow. The next question is then: What are the necessary conditions on the parametrization $G$ so that it enables the gradient flow to simulate a mirror flow? We provide a (partial) characterization of such $G$ in the following theorem.

**Theorem 4.9** (Necessary condition on smooth parametrization to be commuting)**.** *Let $M$ be a smooth submanifold of $\mathbb{R}^D$ and $G : M \to \mathbb{R}^d$ be a smooth parametrization. If for any $x_{\text{init}} \in M$, there is a Legendre function $R$ such that for all time-dependent loss $L_t \in \mathcal{L}$, the gradient flow under $L_t \circ G$ initialized at $x_{\text{init}}$ can be written as the mirror flow under $L_t$ with respect to $R$, then $G$ must be a regular parametrization, and it also holds that for each $x \in M$,*

$$\mathrm{Lie}^{\geq 2}(\partial G)\big|_x \subseteq \ker(\partial G(x)), \tag{9}$$

*where $\mathrm{Lie}^{\geq K}(\partial G) := \mathrm{span}\big\{ [[[[\nabla G_{j_1}, \nabla G_{j_2}], \ldots ], \nabla G_{j_{k-1}}], \nabla G_{j_k}] \mid k \geq K, \forall i \in [k], j_i \in [d] \big\}$ is the subset of the Lie algebra generated by $\{\nabla G_i\}_{i=1}^d$ only containing elements of order higher than $K$, and $\ker(\partial G(x))$ is the orthogonal complement of $\mathrm{span}(\{\nabla G_i(x)\}_{i=1}^d)$ in $\mathbb{R}^D$.*

Note the necessary condition in (9) is weaker than assuming that G is a commuting parametrization, and we conjecture that it is indeed sufficient.

**Conjecture 4.10.** The claim in Theorem 4.8 still holds, if we relax the commuting assumption to that $\mathrm{Lie}^{\geq 2}(\partial G)\big|_x \subseteq \ker(\partial G(x))$ for all $x \in M$.

With the above necessary condition (9), we can formally refute the possibility that one can use mirror flow to characterize the implicit bias of gradient flow for matrix factorization in general settings, as summarized in Corollary 4.11. It is also worth mentioning that [40] constructed a concrete counter example showing that the implicit bias for commuting measurements, that gradient flow finds the solution with minimal nuclear norm, does not hold for the general case, where gradient flow could prefer the solution with minimal rank instead.

**Corollary 4.11** (Gradient flow for matrix factorization cannot be written as mirror flow)**.** *For any $d, r \in \mathbb{N}$, let $M$ be an open set in $\mathbb{R}^{d \times r}$ and $G : M \to \mathbb{R}^{d \times d}$ be a smooth parametrization given by $G(U) = UU^\top$. Then there exists a initial point $U_{init} \in M$ and a time-dependent loss $L_t$ such that the gradient flow under $L_t \circ G$ starting from $U_{init}$ cannot be written as a mirror flow with respect to any Legendre function $R$ under the loss $L_t$.*

The following corollary shows that gradient flow with non-commuting parametrization cannot be mirror flow, when the dimension of the reachable set matches that of the $w$-space.

**Corollary 4.12.** *Let $M$ be a smooth submanifold of $\mathbb{R}^D$ whose dimension is at least $d$. Let $G : M \to \mathbb{R}^d$ be a regular parametrization such that for any $x_{\text{init}} \in M$, (1) $\Omega_x(x_{\text{init}}; G)$ is a submanifold of dimension $d$, and (2) there is a Legendre function $R$ such that for any time-dependent loss $L_t \in \mathcal{L}$, the gradient flow governed by $-\nabla(L_t \circ G)$ with initialization $x_{\text{init}}$ can be written as a mirror flow with respect to $R$. Then $G$ must be a commuting parametrization.*

Next, we establish the convergence of $w(t) = G(x(t))$ when $x(t)$ is given by some gradient flow with the commuting parametrization $G$. Here we require that the convex function $R$ given by Lemma 4.7 is a Bregman function (see definition in Appendix B). The proofs of Theorem 4.13, Corollary 4.14 and Theorem 4.15 are in Appendix D.

**Theorem 4.13.** *Under the setting of Theorem 4.8, further assume that the loss $L$ is quasi-convex, $\nabla L$ is locally Lipschitz and $\mathrm{argmin}\{L(w) \mid w \in \mathrm{dom}\, R\}$ is non-empty where $R : \mathbb{R}^d \to \mathbb{R} \cup \{\infty\}$ is the convex function given by Lemma 4.7. Suppose $R$ is a Bregman function, then as $t \to \infty$, $w(t)$ converges to some $w^*$ such that $\nabla L(w^*)^\top (w - w^*) \geq 0$ for all $w \in \mathrm{dom}\, R$. Moreover, if the loss function $L$ is convex, then $w(t)$ converges to a minimizer in $\overline{\mathrm{dom}\, R}$.*

**Corollary 4.14.** *Under the setting of Theorem 4.13, if the reachable set in the $w$-space satisfies $\Omega_w(x_{\text{init}}; G) = \mathbb{R}^d$, then $R$ is a Bregman function and all the statements in Theorem 4.13 hold.*

**Theorem 4.15.** *Under the setting of Theorem 4.13, consider the commuting quadratic parametrization $G : \mathbb{R}^D \to \mathbb{R}^d$ where each $G_i(x) = \frac{1}{2} x^\top A_i x$, for symmetric matrices $A_1, A_2, \ldots, A_d \in \mathbb{R}^{D \times D}$ that commute with each other, i.e., $A_i A_j - A_j A_i = 0$ for all $i, j \in [d]$. For any $x_{\text{init}} \in \mathbb{R}^D$, if $\{\nabla G_i(x_{\text{init}})\}_{i=1}^d = \{A_i x_{\text{init}}\}_{i=1}^d$ are linearly independent, then the following holds:*

(a) For all $\mu \in \mathbb{R}^d$, $\psi(x_{\text{init}}; \mu) = \exp(\sum_{i=1}^{d} \mu_i A_i) x_{\text{init}}$ where $\exp(\cdot)$ is the matrix exponential defined as $\exp(A) := \sum_{k=0}^{\infty} \frac{A^k}{k!}$.

(b) For each $j \in [d]$ and all $\mu \in \mathbb{R}^d$, $G_j(\psi(x_{\text{init}}; \mu)) = \frac{1}{2} x_{\text{init}}^\top \exp(\sum_{i=1}^{d} 2\mu_i A_i) A_j x_{\text{init}}$.

(c) $Q(\mu) = \frac{1}{4} \|\psi(x_{\text{init}}; \mu)\|_2^2$ is a Legendre function with domain $\mathbb{R}^d$.

(d) $R$ is a Bregman function with $\text{dom}\, R = \overline{\text{range}\, \nabla Q}$ where $\text{range}\, \nabla Q$ is the range of $\nabla Q$, and thus all the statements in Theorem 4.13 hold.

## 4.3 Solving underdetermined linear regression with commuting parametrization

Next, we specialize to underdetermined linear regression problems to showcase our framework.

**Setting: underdetermined linear regression.** Let $\{(z_i, y_i)\}_{i=1}^{n} \subset \mathbb{R}^d \times \mathbb{R}$ be a dataset of size $n$. Given any parametrization $G$, the output of the linear model on the $i$-th data is $z_i^\top G(x)$. The goal is to solve the regression for the label vector $Y = (y_1, y_2, \ldots, y_n)^\top$. For notational convenience, we define $Z = (z_1, z_2, \ldots, z_n) \in \mathbb{R}^{d \times n}$.

We can apply Theorem 3.8 to show the implicit bias of gradient flow with commuting parametrization.

**Theorem 4.16.** *Let $M$ be a smooth submanifold of $\mathbb{R}^d$ and $G : M \to \mathbb{R}^d$ be a commuting and regular parametrization satisfying Assumption 3.5. Suppose the loss function $L$ satisfies $L(w) = \widetilde{L}(Zw)$ for some differentiable $\widetilde{L} : \mathbb{R}^n \to \mathbb{R}$. For any initialization $x_{\text{init}} \in M$, consider the gradient flow*

$$\mathrm{d}x(t) = -\nabla(L \circ G)(x(t))\mathrm{d}t, \qquad x(0) = x_{\text{init}}.$$

*There exists a convex function $R$ (given by Lemma 4.7, depending only on $x_{\text{init}}$ and $G$), such that for any dataset $\{(z_i, y_i)\}_{i=1}^{n} \subset \mathbb{R}^d \times \mathbb{R}$, if $w(t) = G(x(t))$ converges as $t \to \infty$ and the convergence point $w_\infty = \lim_{t \to \infty} w(t)$ satisfies $Zw_\infty = Y$, then $R(w_\infty) = \min_{w : Zw = Y} R(w)$, that is, gradient flow implicitly minimizes the convex regularizer $R$ among all interpolating solutions.*

Note that the identity parametrization $w = G(x) = x$ is a commuting parametrization. Therefore, if we run the ordinary gradient flow on $w$ itself and it converges to some interpolating solution, then the convergence point is closest to the initialization in Euclidean distance among all interpolating solutions. This recovers the well-known implicit bias of gradient flow for underdetermined regression.

Furthermore, we can recover the results on the quadratically overparametrized linear model studied in a series of papers [26, 61, 8], as summarized in the following Corollary 4.17. Note that their results assumed convergence in order to characterize the implicit bias, whereas our framework enables us to directly prove the convergence as in Theorem 4.15. The convergence guarantee here is also more general than existing convergence results for Example 4.3 in [50, 42].

**Corollary 4.17.** *Consider the underdetermined linear regression problem with data $Z \in \mathbb{R}^{d \times n}$ and $Y \in \mathbb{R}^n$. Let $\widetilde{L} : \mathbb{R}^n \to \mathbb{R}$ be a differentiable loss function such that $\widetilde{L}$ is quasi-convex, $\nabla \widetilde{L}$ is locally Lipschitz, and $Y \in \mathbb{R}^n$ is its unique global minimizer. Consider solving $\min_w \widetilde{L}(Zw)$ by running gradient flow on $L(w) = \widetilde{L}(Zw)$ with the quadratic parametrization $w = G(x) = u^{\odot 2} - v^{\odot 2}$ where $x = \binom{u}{v} \in \mathbb{R}_+^{2d}$, for any initialization $x_{\text{init}} \in \mathbb{R}_+^{2d}$: $\mathrm{d}x(t) = -\nabla(L \circ G)(x(t))\mathrm{d}t$, $x(0) = x_{\text{init}}$. Then as $t \to \infty$, $w(t) = G(x(t))$ converges to some $w_\infty$ such that $Zw_\infty = Y$ and $R(w_\infty) = \min_{w : Zw = Y} R(w)$ where $R$ is given by*

$$R(w) = \frac{1}{4} \sum_{i=1}^{d} \left( w_i \,\text{arcsinh}\left(\frac{w_i}{2u_{0,i} v_{0,i}}\right) - \sqrt{w_i^2 + 4u_{0,i}^2 v_{0,i}^2} - w_i \ln \frac{u_{0,i}}{v_{0,i}} \right).$$

# 5 Every mirror flow is a gradient flow with commuting parametrization

For any smooth Legendre function $R : \mathbb{R}^d \to \mathbb{R} \cup \{\infty\}$, recall the corresponding mirror flow:

$$\mathrm{d}\nabla R(w(t)) = -\nabla L(w(t))\mathrm{d}t.$$

Note that $\text{int}(\text{dom}\, R)$ is a convex open set of $\mathbb{R}^d$, hence a smooth manifold (see Example 1.26 in [38]), and $\nabla^2 R$ is a continuous positive-definite metric on $\text{int}(\text{dom}\, R)$. As discussed previously in (3), the above mirror flow is the Riemannian gradient flow on the Riemannian manifold $(\text{int}(\text{dom}\, R), \nabla^2 R)$. The goal is to find a parametrization $G : U \to \mathbb{R}^d$, where $U$ is an open set of $\mathbb{R}^D$, such that the dynamics of $w(t) = G(x(t))$ can be induced by the gradient flow on $x(t)$ governed by $-\nabla(L \circ G)(x)$. Formally, we have the following result:

**Theorem 5.1.** *Let $R : \mathbb{R}^d \to \mathbb{R} \cup \{\infty\}$ be a smooth Legendre function. There exist a smooth submanifold of $\mathbb{R}^D$ denoted by $M$, an open neighborhood $U$ of $M$ and a smooth and regular parametrization $G : U \to \mathbb{R}^d$ such that for the mirror flow under any time-dependent loss $L_t$ with any initialization $w_{\mathrm{init}} \in \mathrm{int}(\mathrm{dom}\, R)$*

$$\mathrm{d}\nabla R(w(t)) = -\nabla L_t(w(t))\mathrm{d}t, \quad w(0) = w_{\mathrm{init}}, \tag{10}$$

*it holds that $w(t) = G(x(t))$ for all $t \geq 0$ where $x(t)$ is given by the gradient flow under $L_t \circ G$:*

$$\mathrm{d}x(t) = -\nabla(L_t \circ G)(x(t))\mathrm{d}t, \quad x(0) = x_{\mathrm{init}} \tag{11}$$

*where $x_{\mathrm{init}}$ satisfies $G(x_{\mathrm{init}}) = w_{\mathrm{init}}$. Moreover, let $G|_M$ be the restriction of $G$ on $M$, then $G|_M$ is a commuting and regular parametrization and $\partial G = \partial G|_M$ on $M$, which implies $x(t) \in M$ for all $t \geq 0$. If $R$ is further a mirror map, then $\{\nabla G_i|_M\}_{i=1}^d$ are complete vector fields on $M$.*

The proof of Theorem 5.1 can be found in Appendix E. To illustrate the idea, let us first suppose such a smooth and regular parametrization $G$ exists and is a bijection between the reachable set $\Omega_x(x_{\mathrm{init}}; G) \subset \mathbb{R}^D$ and $\mathrm{int}(\mathrm{dom}\, R)$, and denote its inverse by $F$. It turns out that we can show

$$\partial F(w)^\top \partial F(w) = (\partial G(F(w))\partial G(F(w))^\top)^{-1} = \nabla^2 R(w)$$

where the second equality follows from the relationship between $R$ and $G$ as discussed in the introduction on (2). Note that this corresponds to expressing the metric tensor $\nabla^2 R$ using an explicit map $F$, which is further equivalent to embedding the Riemannian manifold $(\mathrm{int}(\mathrm{dom}\, R), \nabla^2 R)$ into a Euclidean space $(\mathbb{R}^D, \bar{g})$ in a way that preserves its metric. This refers to a notion called isometric embedding in differential geometry.

**Definition 5.2** (Isometric embedding). Let $(M, g)$ be a Riemannian submanifold of $\mathbb{R}^d$. An *isometric embedding* from $(M, g)$ to $(\mathbb{R}^D, \bar{g})$ is a differentiable injective map $F : M \to \mathbb{R}^D$ that preserves the metric, i.e., for any two tangent vectors $v, w \in T_x M$ it holds that $g_x(v, w) = \bar{g}_x(\partial F(x)v, \partial F(x)w)$.

Nash's embedding theorem is a classic result in differential geometry that guarantees the existence of isometric embedding of any Riemannian manifold into a Euclidean space with a plain geometry. See Appendix A.1 for additional discussion on construction of $G$ given a Legendre function $R$.

**Theorem 5.3** (Nash's embedding theorem, [47, 48, 28]). *Any $d$-dimensional Riemannian manifold has an isometric embedding to $(\mathbb{R}^D, \bar{g})$ for $D = \max\{d(d+5)/2, d(d+3)/2 + 5\}$.*

As another way to understand Theorem 4.8, note that $\nabla^2 R(w)^{-1}\nabla L(w)$ is the Riemannian gradient of $L$ on the Riemannian manifold $(\mathrm{int}(\mathrm{dom}\, R), \nabla^2 R)$. It is well-known that gradient flow is invariant under isometric embedding, and thus we can use Nash's embedding theorem to rewrite the Riemannian gradient flow on $(\mathrm{int}(\mathrm{dom}\, R), g^R)$ as that on $(\mathbb{R}^D, \bar{g})$.

## 6 Conclusion

We presented a framework that characterizes when gradient descent with proper paramterization becomes equivalent to mirror descent. In the limit of infinitesimal step size, we identify a notion named commuting parametrization such that any gradient flow (i.e., the continuous analog of gradient descent) with a commuting parametrization is equivalent to a mirror flow (i.e., the continuous analog of mirror descent) in the original parameter space with respect to a Legendre function that depends only on the initialization and the parametrization. Conversely, we use Nash's embedding theorem to show that any mirror flow can be characterized by a gradient flow in the reparametrized space with a commuting parametrization. Using our framework, we recover and generalize results on the implicit bias of gradient descent in a series of existing works, including a rigorous and general proof of convergence. We also provide a necessary condition for the parametrization such that gradient flow in the reparametrized space is equivalent to a mirror flow in the original space. However, the necessary condition is slightly weaker than the commuting condition and it is left for future work to close the gap.

## Acknowledgement

This work was supported by NSF, DARPA/SRC, Simons Foundation, and ONR. ZL acknowledges support of Microsoft Research PhD Fellowship and JDL acknowledges support of the ARO under MURI Award W911NF-11-1-0304, the Sloan Research Fellowship, NSF CCF 2002272, NSF IIS 2107304, ONR Young Investigator Award, and NSF CAREER Award 2144994.

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
