# A  Additional Results

We provide additional results summarized as follows. We discuss how to construct the parametrization $G$ from a given Legendre function $R$ in Appendix A.1. We discuss the existence of non-separable commuting parametrization in Appendix A.2.

## A.1  Examples of constructing $G$ from a given Legendre function $R$

Since the construction of the isometric embedding in Nash's embedding theorem is not explicit and infeasible to compute in general, the corresponding parametrization $G$ given by Theorem 5.1 does not admit an analytic formula for general mirror map $R$. However, in many cases involving standard convex functions $R$ which are separable, it is indeed tractable to explicitly compute the corresponding parametrization $G$.

For example, for Burg entropy ($R(w) = -\sum_{i=1}^{d} \log w_i$) and negative entropy ($R(w) = \sum_{i=1}^{d} w_i \log w_i$), it is easy to verify that we can choose $w = G(x) = (e^{x_1}, \ldots, e^{x_d})$ and $w = G(x) = (x_1^2/4, \ldots, x_d^2/4)$ respectively, where in both cases $x$ has the same dimension as $w$ does. (See also Example 2 and 4 in [5].) More generally, suppose $R$ satisfies that $\nabla^2 R(w)$ is always diagonal, it suffices to find a $F_i : \mathbb{R} \to \mathbb{R}$ such that $F_i'(w_i) = \sqrt{\partial_{ii} R(w_i)}, \forall w_i \in \mathbb{R}$, which can be solved easily by integral. Once we have $F_i$, the parametrization $G : \mathbb{R}^d \to \mathbb{R}^d$ defined by $G_i = F_i^{-1}$ is the desired commuting parametrization with respect to which gradient flow can be written as mirror flow with respect to $R$. (Note $G_i$ is well-defined because $F_i$ is monotone increasing) This is because $\partial G(G^{-1}(w)) \partial G(G^{-1}(w))^\top = (\partial G^{-1}(w) \partial G^{-1}(w)^\top)^{-1} = (\nabla^2 R(w))^{-1}$.

Finally, we also want to remark that in our application of Nash's embedding theorem, the Riemannian metric is given by the Hessian of a mirror map, and it is not clear if this would endow a more explicit and tractable construction of the isometric embedding. We are not aware of such results to the best of our knowledge.

## A.2  Existence of non-separable commuting parametrization

Despite the recent line of works on the connection between mirror descent and gradient descent [24, 4, 5, 8, 23], so far we have not seen any concrete example of non-separable parametrization (in the sense of Definition A.1) such that the reparametrized gradient flow can be written as a mirror flow. In this subsection, we discuss how we can use Theorem 5.1 to construct non-separable, yet commuting parametrizations.

**Definition A.1** (Generalized separable parametrization). Let $M$ be an open subset of $\mathbb{R}^D$. We say a function $G : M \to \mathbb{R}^d$ is a *generalized separable parametrization* if and only if there exist $d$ projection matrices $\{P_i\}_{i=1}^{d}$ satisfying $\sum_{i=1}^{d} P_i = I_d$, $P_i P_j = \mathbb{1}\{i = j\} \cdot P_i$, a function $\widehat{G} : M \to \mathbb{R}^d$ satisfying $\widehat{G}_i(x) = \widehat{G}_i(P_i x)$, a matrix $A \in \mathbb{R}^{d \times d}$ and a vector $b \in \mathbb{R}^d$, such that

$$G(x) = A\widehat{G}(x) + b, \qquad \forall x \in M.$$

Given the above definition, it is easy to check that $\widehat{G}$ is a commuting parametrization as $\nabla^2 \widehat{G}_i \nabla \widehat{G}_j = P_i \nabla^2 \widehat{G}_i P_i \cdot P_j \nabla \widehat{G}_j \equiv 0$ for all $i \neq j$, so each Lie bracket $[\nabla G_i, \nabla G_j]$ is also 0 by the linearity.

As a concrete example, for matrix sensing with commutable measurement $A_1, \ldots, A_m \in \mathbb{R}^{d \times d}$ (see Example 4.4 and Remark 4.5), let $V = (v_1, \ldots, v_d) \in \mathbb{R}^{d \times d}$ be a common eigenvector matrix for $\{A_i\}_{i=1}^{m}$ such that we can write $A_i = V \Sigma_i V^\top = \sum_{j=1}^{d} \sigma_{i,j} v_j v_j^\top$ for each $i \in [m]$. For parametrization $G : \mathbb{R}^{d \times r} \to d$ where each $G_i(U) = v_i^\top U U^\top v_i$, we can write $\langle A_i, UU^\top \rangle = \sum_{j=1}^{d} \sigma_{i,j} G_j(U)$.

However, the bad news is that separable commuting parametrizations can express only a restricted class of Legendre functions. It is easy to see $\partial \hat{G}(x) \partial \hat{G}(x)^\top$ must be diagonal for every $x$. Thus $\partial G(x) \partial G(x)^\top$ is simultaneously diagonalizable for all $x$, and so is the Hessian of the corresponding Legendre function (given by Lemma 4.7). Yet there are interesting Legendre functions whose

Hessians are not simultaneously diagonalizable, such as

$$R(w) = \sum_{i=1}^{d} w_i(\ln w_i - 1) + \left(1 - \sum_{i=1}^{d} w_i\right)\left(\ln\left(1 - \sum_{i=1}^{d} w_i\right) - 1\right),$$

where each $w_i > 0$ and $\sum_{i=1}^{d} w_i < 1$. We can check that $\nabla R(w) = \sum_{i=1}^{d} \ln \frac{w_i}{1 - \sum_{i=1}^{d} w_i}$ and $\nabla^2 R(w) = \mathrm{diag}(w^{\odot(-1)}) + \mathbb{1}_d\mathbb{1}_d^\top$. Indeed, it is proposed as an open problem by [5] whether we can find a parametrization $G$ such that the reparametrized gradient flow in the $x$-space simulates the mirror flow in the $w$-space with respect to the aforementioned Legendre function $R$.

Our Theorem 5.1 answers the open problem by [5] affirmatively since it shows every mirror flow can be written as some reparametrized gradient flow. According to the previous discussion, every mirror flow for Legendre function whose Hessian cannot be simultaneously diagonalized always induces a non-separable commuting parametrization. But this type of construction has two caveats: First, the construction of the Legendre function uses Nash's Embedding theorem, which is implicit and hard to implement; second, the parametrization given by Theorem 5.1, though defined on an open set in $\mathbb{R}^D$, is only commuting on the reachable set, which is a $d$-dimensional submanifold of $\mathbb{R}^D$. This is different from all the natural examples of commuting parametrizations that are commuting on an open set, leading to the following open question.

**Open Question:** Is there any smooth, regular, commuting, yet non-separable (in the sense of Definition A.1) parametrization from an open subset of $\mathbb{R}^D$ to $\mathbb{R}^d$, for some integers $D$ and $d$?

**Theorem A.2.** *All smooth, regular and commuting parametrizations are non-separable when $D = 1$.*

*Proof of Theorem A.2.* Note that $[\nabla G_i, \nabla G_j] \equiv 0$ implies that all $G_i$ share the same set of stationary points, *i.e.*, $\{x \in \mathbb{R} \mid \nabla G_i(x) = 0\}$ is the same for all $i \in [d]$. Since $D = 1$, without loss of generality, we can assume $G_i'(x) = \nabla G_i(x) > 0$ for all $x \in M$ and $i \in [d]$ since $G$ is regular. Then it holds that $\mathrm{sign}(G_i')(\ln |G_i'|)' = \mathrm{sign}(G_j')(\ln |G_j'|)'$, which implies that $|G_i'|/|G_j'|$ is equal to some constant independent of $x$. This completes the proof. $\square$

**Remark A.3.** *We note that the assumption that the parametrization is regular is necessary for the open question to be non-trivial. Otherwise, consider the following example with $D = 1$ and $d = 2$: Let $f_1, f_2 : \mathbb{R} \to \mathbb{R}$ be any smooth function supported on $(0, 1)$ and $(1, 2)$ respectively. Define $G_i(x) = \int_0^x f_i(t)\mathrm{d}t$ for all $x \in \mathbb{R}$. Then parametrization $G$ is non-separable.*

## B  Related basics for convex analysis

We first introduce some additional notations. For any function $f$, we denote its range (or image) by $\mathrm{range}\, f$. For any set $S$, we use $\overline{S}$ to denote its closure. For any matrix $\Lambda \in \mathbb{R}^{d \times D}$ and set $S \subseteq \mathbb{R}^D$, we define $\Lambda S = \{\Lambda x \mid x \in S\} \subseteq \mathbb{R}^d$.

Below we collect some related basic definitions and results in convex analysis. We refer the reader to [54] and [9] as main reference sources. In particular, Sections 2, 3 and 4 in [9] provide a clear summary of the related concepts.

Here we consider a convex function $f : \mathbb{R}^d \to \mathbb{R} \cup \{\infty\}$ whose domain is $\mathrm{dom}\, f = \{w \in \mathbb{R}^d \mid f(w) < \infty\}$. **From now on, we assume by default** that $f$ is continuous on $\mathrm{dom}\, f$, the interior of its domain $\mathrm{int}(\mathrm{dom}\, f)$ is non-empty, and $f$ is differentiable on $\mathrm{int}(\mathrm{dom}\, f)$.

The notions of essential smoothness and essential strict convexity defined below describe certain nice properties of a convex function (see Section 26 in [54]).

**Definition B.1** (Essential smoothness and essential strict convexity)**.** If for any sequence $\{w_n\}_{n=1}^{\infty} \subset \mathrm{int}(\mathrm{dom}\, f)$ going to the boundary of $\mathrm{dom}\, f$ as $n \to \infty$, it holds that $\|\nabla f(w_n)\| \to \infty$, then we say $f$ is *essentially smooth*. If $f$ is strictly convex on every convex subset of $\mathrm{int}(\mathrm{dom}\, f)$, then we say $f$ is *essentially strictly convex*.

The concept of *convex conjugate* is critical in our derivation. Specifically, given a convex function $f : \mathbb{R}^d \to \mathbb{R} \cup \{\infty\}$, its convex conjugate $f^*$ is defined as

$$f^*(w) = \sup_{y \in \mathbb{R}^d} \langle w, y \rangle - f(y).$$

The following results characterize the relationship between a convex function and its conjugate.

**Theorem B.2** (Theorem 26.3, [54])**.** *A convex function $f$ is essentially strictly convex if and only if its convex conjugate $f^*$ is essentially smooth.*

**Proposition B.3** (Proposition 2.5, [9])**.** *If $f$ is essentially strictly convex, then $\mathrm{range}\,\partial f = \mathrm{int}(\mathrm{dom}\,f^*) = \mathrm{dom}\,\nabla f^*$, where $\partial f$ is the subgradient of $f$.*

**Lemma B.4** (Corollary 2.6, [9])**.** *If $f$ is essentially strictly convex, then it holds for all $w \in \mathrm{int}(\mathrm{dom}\,f)$ that $\nabla f(w) \in \mathrm{int}(\mathrm{dom}\,f^*)$ and $\nabla f^*(\nabla f(w)) = w$.*

The class of Legendre functions defined in Definition 3.7 contains convex functions that are both essentially smooth and essentially strictly convex.

**Theorem B.5** (Theorem 26.5, [54])**.** *A convex function $f$ is a Legendre function if and only if its conjugate $f^*$ is. In this case, the gradient mapping $\nabla f : \mathrm{int}(\mathrm{dom}\,f) \rightarrow \mathrm{int}(\mathrm{dom}\,f^*)$ satisfies $(\nabla f)^{-1} = \nabla f^*$.*

Next, we introduce the notion of Bregman function [12, 14]. It has been shown in [9] that the properties of Bregman functions are crucial to prove the trajectory convergence of Riemannian gradient flow where the metric tensor is given by the Hessian of some Bregman function $f$.

**Definition B.6** (Bregman functions; Definition 4.1, [3])**.** A function $f$ is called a *Bregman function* if it satisfies the following properties:

(a) $\mathrm{dom}\,f$ is closed. $f$ is strictly convex and continuous on $\mathrm{dom}\,f$. $f$ is $\mathcal{C}^1$ on $\mathrm{int}(\mathrm{dom}\,f)$.

(b) For any $w \in \mathrm{dom}\,f$ and $\alpha \in \mathbb{R}$, $\{y \in \mathrm{dom}\,f \mid D_R(w, y) \leq \alpha\}$ is bounded.

(c) For any $w \in \mathrm{dom}\,f$ and sequence $\{w_i\}_{i=1}^\infty \subset \mathrm{int}(\mathrm{dom}\,f)$ such that $\lim_{i\to\infty} w_i = w$, it holds that $\lim_{i\to\infty} D_R(w, w_i) \to 0$.

The following theorem provides a special sufficient condition for $f$ to be a Bregman function.

**Theorem B.7** (Theorem 4.7, [3])**.** *If $f$ is a Legendre function with $\mathrm{dom}\,f = \mathbb{R}^d$, then $\mathrm{dom}\,f^* = \mathbb{R}^d$ implies that $f$ is a Bregman function.*

The following theorem from [3] provides a convenient tool for proving the convergence of a Riemannian gradient flow.

**Theorem B.8** (Theorem 4.2, [3])**.** *Suppose $f : \mathbb{R}^d \to \mathbb{R} \cup \{\infty\}$ is a Bregman function and also a Legendre function, and satisfies that $f$ is twice continuously differentiable on $\mathrm{int}(\mathrm{dom}\,f)$ and $\nabla^2 f$ is locally Lipschitz. Consider the following Riemannian gradient flow:*

$$\mathrm{d}w(t) = -\nabla^2 f(w(t))^{-1} \nabla L(w(t))\mathrm{d}t, \qquad w(0) = w_{\mathrm{init}} \in \mathrm{int}(\mathrm{dom}\,f)$$

*where the loss $L : \mathbb{R}^d \to \mathbb{R}$ satisfies that $L$ is quasi-convex, $\nabla L$ is locally Lipschitz, and $\mathrm{argmin}\{L(w) \mid w \in \mathrm{dom}\,f\}$ is non-empty. Then as $t \to \infty$, $w(t)$ converges to some $w^* \in \mathrm{dom}\,f$ such that $\langle \nabla L(w^*), w - w^* \rangle \geq 0$ for all $w \in \mathrm{dom}\,f$. If the loss $L$ is further convex, then $w^*$ is a minimizer of $L$ on $\mathrm{dom}\,f$.*

## C   Omitted proofs in Section 3

Here we first present the result and its proof for the domain of the flow induced by $G$.

**Lemma C.1.** *Let $M$ be a smooth submanifold of $\mathbb{R}^D$ and $G : M \to \mathbb{R}^d$ be a $\mathcal{C}^2$ parametrization satisfying Assumption 3.5. Then for any $x \in M$, $\mathcal{U}(x)$ is a hyperrectangle, i.e., $\mathcal{U}(x)$ can be decomposed as*

$$\mathcal{U}(x) = \mathcal{I}_1(x) \times \mathcal{I}_2(x) \times \cdots \times \mathcal{I}_d(x)$$

*where $\mathcal{I}_j(x) := \{x'_j \mid x' \in \mathcal{U}(x)\}$ is an open interval.*

*Proof of Lemma C.1.* Fix any $x \in M$. For each $i \in [d]$, let $\mathcal{I}_i(x)$ be the domain of $\phi_{G_j}^t(x)$ in terms of $t$. If $\nabla G_i$ is a complete vector field on $M$ as in Definition 3.2, then $\mathcal{I}_i(x) = \mathbb{R}^d$, otherwise $\phi_{G_j}^t(x)$ is defined for $t$ in an open interval containing 0 (see, e.g., Theorem 2.1 in [37]). Then we claim

that for any distinct $j_1, j_2, \ldots, j_k \in [d]$ where $k \in [d]$, the set of all $(\mu_{j_1}, \ldots, \mu_{j_k}) \in \mathbb{R}^k$ such that $\phi_{G_{j_1}}^{\mu_{j_1}} \circ \cdots \circ \phi_{G_{j_k}}^{\mu_{j_k}}(x)$ is well-defined is a hyperrectangle given by $\mathcal{I}_{j_1}(x) \times \mathcal{I}_{j_2}(x) \times \cdots \times \mathcal{I}_{j_k}(x)$. Then the desired result can be obtained by letting $(j_1, j_2, \ldots, j_d) = (1, 2, \ldots, d)$. We prove the claim by induction over $k \in [d]$.

The base case for $k = 1$ has already been established above. Next, assume the claim holds for $1, 2, \ldots, k-1$ where $k \geq 3$, and we proceed to show it for $k$. By the claim for $k-2$, $\phi_{G_{j_3}}^{\mu_{j_3}} \circ \cdots \circ \phi_{G_{j_k}}^{\mu_{j_k}}(x)$ is well-defined for $(\mu_{j_3}, \ldots, \mu_{j_k}) \in \mathcal{I}_{j_3}(x) \times \cdots \times \mathcal{I}_{j_k}(x)$. For any such $(\mu_{j_3}, \ldots, \mu_{j_k})$, $\phi_{G_{j_1}}^t \circ \phi_{G_{j_3}}^{\mu_3} \circ \cdots \circ \phi_{G_{j_k}}^{\mu_{j_k}}(x)$ is well-defined for $t$ in and only in the open interval $\mathcal{I}_{j_1}(x)$ by applying the claim for $k-1$, and similarly $\phi_{G_{j_2}}^t \circ \phi_{G_{j_3}}^{\mu_3} \circ \cdots \circ \phi_{G_{j_k}}^{\mu_{j_k}}(x)$ is also well-defined for $t$ in and only in the open interval $\mathcal{I}_{j_2}(x)$. Note that for any $(s, t) \in \mathcal{I}_{j_1}(x) \times \mathcal{I}_{j_2}(x)$,

$$\phi_{G_{j_1}}^s \circ \phi_{G_{j_2}}^{-t} \circ \phi_{G_{j_2}}^t \circ \phi_{G_{j_3}}^{\mu_{j_3}} \circ \cdots \circ \phi_{G_{j_k}}^{\mu_{j_k}}(x)$$

is well-defined, so by Assumption 3.5, we see that

$$\phi_{G_{j_2}}^{-t} \circ \phi_{G_{j_1}}^s \circ \phi_{G_{j_2}}^t \circ \phi_{G_{j_3}}^{\mu_{j_3}} \circ \cdots \circ \phi_{G_{j_k}}^{\mu_{j_k}}(x)$$

is also well-defined, which further implies that $\phi_{G_{j_1}}^s \circ \phi_{G_{j_2}}^t \circ \phi_{G_{j_3}}^{\mu_{j_3}} \circ \cdots \circ \phi_{G_{j_k}}^{\mu_{j_k}}(x)$ is well-defined. Therefore, we conclude that $\phi_{G_{j_1}}^{\mu_{j_1}} \circ \cdots \circ \phi_{G_{j_k}}^{\mu_{j_k}}(x)$ is well-defined for and only for $(\mu_{j_1}, \ldots, \mu_{j_k}) \in \mathcal{I}_{j_1}(x) \times \cdots \times \mathcal{I}_{j_k}(x)$. This completes the induction and hence finishes the proof. $\square$

Next, we provide the proof for the implicit bias of mirror flow summarized in Theorem 3.8. We need the following lemma that characterizes the KKT conditions for minimizing a convex function $R$ in a linear subspace.

**Lemma C.2.** *For any convex function $R : \mathbb{R}^d \to \mathbb{R} \cup \{\infty\}$ and $Z \in \mathbb{R}^{n \times d}$, suppose $\nabla R(w^*) = Z^\top \lambda$ for some $\lambda \in \mathbb{R}^n$, then*

$$R(w^*) = \min_{w : Z(w - w^*) = 0} R(w).$$

*Proof of Lemma C.2.* Consider another convex function defined as $\widetilde{R}(w) = R(w) - w^\top Z^\top \lambda$, then $\nabla \widetilde{R}(w^*) = \nabla R(w^*) - Z^\top \lambda = 0$, which implies that

$$
\begin{aligned}
\tilde{R}(w^*) &= \min_{w \in \mathbb{R}^d} R(w) - w^\top Z^\top \lambda \\
&\leq \min_{w : Z(w - w^*) = 0} R(w) - w^\top Z^\top \lambda \\
&= \min_{w : Z(w - w^*) = 0} R(w) - w^{*\top} Z^\top \lambda.
\end{aligned}
$$

Since $\widetilde{R}(w^*) = R(w^*) - w^{*\top} Z^\top \lambda$, it follows that

$$R(w^*) \leq \min_{w : Z(w - w^*) = 0} R(w),$$

and the equality is achieved at $w = w^*$. This finishes the proof. $\square$

We can then prove Theorem 3.8 by using Lemma C.2.

*Proof of Theorem 3.8.* Since $L(w) = \widetilde{L}(Zw - Y)$, the mirror flow (8) can be further written as

$$\mathrm{d}\nabla R(w(t)) = -Z^\top \nabla \widetilde{L}(Zw(t) - Y)\mathrm{d}t.$$

Integrating the above yields that for any $t \geq 0$,

$$\nabla R(w(t)) - \nabla R(w_0) = -Z^\top \int_0^t \nabla \widetilde{L}(Zw(s) - Y)\mathrm{d}s \in \mathrm{span}(X^\top),$$

which further implies that $\nabla R(w_\infty) - \nabla R(w_0) \in \mathrm{span}(Z^\top)$. Therefore,

$$\nabla D_R(w, w_0)|_{w = w_\infty} = \nabla R(w_\infty) - \nabla R(w_0) \in \mathrm{span}(Z^\top).$$

Then applying Lemma C.2 yields

$$D_R(w_\infty, w_0) = \min_{w : Z(w - w_\infty) = 0} D_R(w, w_0).$$

This finishes the proof. $\square$

# D Omitted proofs in Section 4

Here we provide the omitted proofs in Section 4, including four main parts:

(1) Properties of commuting parametrizations (Appendix D.1);

(2) Necessary condition for a smooth parametrization to be commuting (Appendix D.2);

(3) Convergence for gradient flow with commuting parametrization (Appendix D.3);

(4) Results for the underdetermined linear regression (Appendix D.4).

## D.1 Properties of commuting parametrizations

We first recall the following result on the characterization of commuting vector fields from [38].

**Theorem D.1** (Adapted from Theorem 9.44 in [38]). *Let $M$ be a smooth submanifold of $\mathbb{R}^D$ and $G : M \to \mathbb{R}^d$ be a $\mathcal{C}^2$ parametrization. For any $i, j \in [d]$, $[\nabla G_i, \nabla G_j](x) = 0$ for all $x \in M$ if and only if for any $x \in M$, whenever both $\phi^s_{G_i} \circ \phi^t_{G_j}(x)$ and $\phi^t_{G_j} \circ \phi^s_{G_i}(x)$ are well-defined for all $(s, t)$ in some rectangle $\mathcal{I}_1 \times \mathcal{I}_2$ where $\mathcal{I}_1, \mathcal{I}_2 \subseteq \mathbb{R}$ are open intervals, it holds that $\phi^s_{G_i} \circ \phi^t_{G_j}(x) = \phi^t_{G_j} \circ \phi^s_{G_i}(x)$ for all $(s, t) \in \mathcal{I}_1 \times \mathcal{I}_2$.*

*Proof of Theorem 4.2.* Note that under Assumption 3.5, Lemma C.1 implies that the domain of $\phi^s_{G_i} \circ \phi^t_{G_j}(x)$ is exactly $\mathcal{I}_i(x) \times \mathcal{I}_j(x)$, and the statement of Theorem 4.2 immediately follows. $\square$

Next, we prove the representation formula for gradient flow with commuting parametrization given in Lemma 4.6.

*Proof of Lemma 4.6.* Let $\mu(t)$ be given by the following differential equation:

$$\mathrm{d}\mu(t) = -\nabla L_t(G(\psi(x_{\mathrm{init}}; \mu(t))))\mathrm{d}t, \qquad \mu(0) = 0.$$

For any $\mu \in \mathcal{U}(x)$ and $j \in [d]$, $\mu + \delta e_j \in \mathcal{U}(x)$ for all sufficiently small $\delta$, thus

$$
\begin{aligned}
\frac{\partial}{\partial \mu_j} \psi(x_{\mathrm{init}}; \mu) &= \lim_{\delta \to 0} \frac{\psi(x_{\mathrm{init}}; \mu + \delta e_j) - \psi(x_{\mathrm{init}}; \mu)}{\delta} \\
&= \lim_{\delta \to 0} \frac{\phi^\delta_{G_j}(\psi(x_{\mathrm{init}}; \mu)) - \psi(x_{\mathrm{init}}; \mu)}{\delta} \\
&= \nabla G_j(\psi(x_{\mathrm{init}}; \mu))
\end{aligned}
$$

where the second equality follows from the assumption that $G$ is a commuting parametrization and Theorem 4.2. Then we have $\frac{\partial \psi(x_{\mathrm{init}}; \mu)}{\partial \mu} = \partial G(\psi(x_{\mathrm{init}}; \mu))^\top$ for all $\mu \in \mathcal{U}(x_{\mathrm{init}})$, and thus when $\mu(t) \in \mathcal{U}(x_{\mathrm{init}})$,

$$
\begin{aligned}
\mathrm{d}\psi(x_{\mathrm{init}}; \mu(t)) &= \frac{\partial \psi(x_{\mathrm{init}}; \mu(t))}{\partial \mu(t)} \mathrm{d}\mu(t) \\
&= -\partial G(x_{\mathrm{init}}; \mu(t)) \nabla L_t(G(\psi(x_{\mathrm{init}}; \mu(t)))) \mathrm{d}t \\
&= -\nabla(L_t \circ G)(\psi(x_{\mathrm{init}}; \mu(t))) \mathrm{d}t.
\end{aligned}
$$

Then since $\psi(x_{\mathrm{init}}; \mu(0)) = x_{\mathrm{init}}$ and $\psi(x_{\mathrm{init}}; \mu(t))$ follows the same differential equation and has the same initialization as $x(t)$, we have $x(t) \equiv \psi(x_{\mathrm{init}}; \mu(t))$ for all $t \in [0, T)$. Therefore,

$$\mu(t) = \mu(0) + \int_0^t -\nabla L_t(G(\psi(x_{\mathrm{init}}; \mu(s)))) \mathrm{d}s = \int_0^t -\nabla L_t(G(x(s))) \mathrm{d}s$$

for all $t \in [0, T)$, which completes the proof. $\square$

Next, to prove Lemma 4.7, we need the following lemma which provides a sufficient condition for a vector function to be gradient of some other function.

**Lemma D.2.** *Let $\Psi : C \to \mathbb{R}^d$ be a differentiable function where $C$ is a simply connected open subset of $\mathbb{R}^d$. If for all $w \in C$ and any $i, j \in [d]$, $\frac{\partial}{\partial w_j} \Psi_i(w) = \frac{\partial}{\partial w_i} \Psi_j(w)$, then there exists some function $Q : C \to \mathbb{R}$ such that $\Psi = \nabla Q$.*

*Proof of Lemma D.2.* This follows from a direct application of Corollary 16.27 in [38]. □

Based on the above results, we proceed to prove Lemma 4.7.

*Proof of Lemma 4.7.* By Lemma C.1, $\mathcal{U}(x_{\text{init}})$ is hyperrectangle, and hence is convex. Next, recall that by the proof of Lemma 4.6, we have $\frac{\partial \psi(x_{\text{init}};\mu)}{\partial \mu} = \partial G(\psi(x_{\text{init}};\mu))^\top$ for all $\mu \in \mathcal{U}(x_{\text{init}})$. Denoting $\Psi(\mu) = G(\psi(x_{\text{init}};\mu))$, we further have

$$\partial \Psi(\mu) = \frac{\partial G(\psi(x_{\text{init}};\mu))}{\partial \psi(x_{\text{init}};\mu)} \frac{\partial \psi(x_{\text{init}};\mu)}{\partial \mu} = \partial G(\psi(x_{\text{init}};\mu)) \partial G(\psi(x_{\text{init}};\mu))^\top, \quad \forall \mu \in \mathcal{U}(x).$$

Since $G$ is regular, $\partial G(\psi(x_{\text{init}};\mu))$ is of full-rank for all $\mu \in \mathcal{U}(x_{\text{init}})$, so $\partial \Psi$ is symmetric and positive definite for all $\mu \in \mathcal{U}(x_{\text{init}})$, which implies that $\Psi$ is the gradient of some strictly convex function $Q : \mathbb{R}^d \to \mathbb{R} \cup \{\infty\}$ by Lemma D.2. This $Q$ satisfies that $\nabla Q(\mu) = \Psi(\mu) = G(\psi(x_{\text{init}};\mu))$ for all $\mu \in \mathcal{U}(x_{\text{init}})$. Therefore, $Q$ is a strictly convex function with $\text{dom}\,\nabla Q = \mathcal{U}(x_{\text{init}})$ and range $\nabla Q = \Omega_w(x_{\text{init}};G)$.

Next, we show that $Q$ is essentially smooth. If $\mathcal{U}(x_{\text{init}}) = \mathbb{R}^d$, then $\text{dom}\,Q = \mathbb{R}^d$ and the boundary of $\text{dom}\,Q$ is empty, so it is trivial that $Q$ is essentially smooth. Otherwise, it suffices to show that for any $\mu$ on the boundary of $\text{dom}\,Q$ and any sequence $\{\mu_k\}_{k=1}^\infty \subset \mathcal{U}(x_{\text{init}})$ such that $\lim_{k\to\infty} \mu_k = \mu_\infty$, we have $\lim_{k\to\infty} \|\nabla Q(\mu_k)\|_2 = \infty$. Since each $\nabla Q(\mu_k) = G(\psi(x_{\text{init}};\mu_k))$, we only need to show that $\lim_{k\to\infty} \|G(\psi(x_{\text{init}};\mu_k))\|_2 = \infty$. Suppose otherwise, then $\{G(\psi(x_{\text{init}};\mu_k)\}_{k=1}^\infty$ is bounded. Note that by Lemma 4.6, let $H_k(x) = \langle \mu_k, G(x) \rangle$, and we have

$$\psi(x_{\text{init}};\mu_k) = \phi^1_{-H_k}(x_{\text{init}}) = x_{\text{init}} + \int_0^1 \nabla H_k(\phi^s_{-H_k}(x_{\text{init}})) \mathrm{d}s.$$

Therefore,

$$\|\psi(x_{\text{init}};\mu_k) - x_{\text{init}}\|_2 \le \int_0^1 \left\|\nabla H_k(\phi^s_{-H_k}(x_{\text{init}}))\right\|_2 \mathrm{d}s \le \sqrt{\int_0^1 \left\|\nabla H_k(\phi^s_{-H_k}(x_{\text{init}}))\right\|_2^2 \mathrm{d}s}. \tag{12}$$

where the second inequality follows from Cauchy-Schwarz inequality. Further note that

$$\begin{aligned}
H_k(\psi(x_{\text{init}};\mu_k)) - H_k(x_{\text{init}}) &= \int_0^1 \frac{\mathrm{d}}{\mathrm{d}s} H_k(\phi^s_{-H_k}(x_{\text{init}})) \mathrm{d}s \\
&= \int_0^1 \left\langle \nabla H_k(\phi^s_{-H_k}(x_{\text{init}})), \frac{\mathrm{d}\phi^s_{-H_k}(x_{\text{init}})}{\mathrm{d}s} \right\rangle \mathrm{d}s \\
&= \int_0^1 \|\nabla H_k(\phi^s_{-H_k}(x_{\text{init}}))\|_2^2 \mathrm{d}s. 
\end{aligned} \tag{13}$$

Then combining (12) and (13), we get

$$\begin{aligned}
\|\psi(x_{\text{init}};\mu_k) - x_{\text{init}}\|_2 &\le \sqrt{\langle \mu_k, G(\psi(x_{\text{init}};\mu_k)) - G(x_{\text{init}}) \rangle} \\
&\le \sqrt{\|\mu_k\|_2 \cdot \|G(\psi(x_{\text{init}};\mu_k)) - G(x_{\text{init}})\|_2},
\end{aligned}$$

which implies that $\{\psi(x_{\text{init}};\mu_k)\}_{k=1}^\infty$ is bounded. Then there exists a convergent subsequence of $\{\psi(x_{\text{init}};\mu_k)\}_{k=1}^\infty$, and without loss of generality we assume that $\psi(x_{\text{init}};\mu_k)$ itself converges to some $x_\infty \in M$ as $k \to \infty$. Note that $\psi(x_\infty;\mu)$ is well-defined for $\mu$ in a small open neighborhood of $0$, and since $\lim_{k\to\infty} \psi(x_{\text{init}};\mu_k) = x_\infty$, for sufficiently large $k$, $\psi(\psi(x_{\text{init}};\mu_k);\mu)$ is well-defined for $\mu$ in a small neighborhood of $0$ that does not depend on $k$. Thus there exists some $\mu \in \mathbb{R}^d$ such that $\mu_k + \mu \notin \mathcal{U}(x_{\text{init}})$ but $\psi(\psi(x_{\text{init}};\mu_k);\mu)$ is well-defined for sufficiently large $k$. But by Lemma C.1 and Theorem D.1, $\psi(\psi(x_{\text{init}};\mu_k);\mu) = \psi(x_{\text{init}};\mu_k + \mu)$ and thus $\mu_k + \mu \in \mathcal{U}(x_{\text{init}})$, which leads to a contradiction. Hence, we conclude that $Q$ is essentially smooth.

Combining the above, it follows that $Q$ is a Legendre function. Let $R : \mathbb{R}^d \to \mathbb{R} \cup \{\infty\}$ be the convex conjugate of $Q$. Then by Theorem B.5, $R$ is also a Legendre function. Note that for any $\mu \in \mathcal{U}(x_{\text{init}})$, by the result in [17], we have

$$\nabla^2 R(G(\psi(x_{\text{init}};\mu))) = \nabla^2 R(\nabla Q(\mu)) = \nabla^2 Q(\mu)^{-1} = (\partial G(\psi(x_{\text{init}};\mu)) \partial G(\psi(x_{\text{init}};\mu))^\top)^{-1}.$$

Therefore, $R$ and $Q$ are both Legendre functions, and by Proposition B.3, we further have range $\nabla R = \operatorname{int}(\operatorname{dom} Q) = \operatorname{dom} \nabla Q = \mathcal{U}(x)$ and conversely $\operatorname{dom} \nabla R = \operatorname{range} \nabla Q = \Omega_w(x_{\text{init}}; G)$. This finishes the proof. $\qquad\square$

Then using Lemma 4.6 and Lemma 4.7, we can prove Theorem 4.8.

*Proof of Theorem 4.8.* Recall that the gradient flow in the $x$-space governed by $-\nabla(L_t \circ G)(x)$ is

$$\mathrm{d}x(t) = -\nabla(L_t \circ G)(x(t))\mathrm{d}t = -\partial G(x(t))^\top \nabla L_t(G(x(t)))\mathrm{d}t.$$

Using $w(t) = G(x(t))$, the corresponding dynamics in the $w$-space is

$$\mathrm{d}w(t) = \partial G(x(t))\mathrm{d}x(t) = -\partial G(x(t))\partial G(x(t))^\top \nabla L_t(w(t))\mathrm{d}t. \tag{14}$$

By Lemma 4.6, we know that the solution to the gradient flow satisfies $x(t) = \psi(x_{\text{init}}; \mu(t))$ where $\mu(t) = \int_0^t -\nabla L_t(G(x(s)))\mathrm{d}s$. Therefore, applying Lemma 4.7, we get a Legendre function $R: \mathbb{R}^d \to \mathbb{R} \cup \{\infty\}$ with domain $\Omega_w(x_{\text{init}}; G)$ such that

$$\nabla^2 R(w(t)) = \nabla^2 R(G(\psi(x_{\text{init}}; \mu(t)))) = \left(\partial G(\psi(x_{\text{init}}; \mu(t)))\partial G(\psi(x_{\text{init}}; \mu(t)))\right)^{-1}$$

for all $t \geq 0$. Then the dynamics of $w(t)$ in (14) can be rewritten as

$$\mathrm{d}w(t) = -\nabla^2 R(w(t))^{-1}\nabla L_t(w(t))\mathrm{d}t,$$

or equivalently,

$$\mathrm{d}\nabla R(w(t)) = -\nabla L_t(w(t))\mathrm{d}t,$$

which is exactly the mirror flow with respect to $R$ initialized at $w(0) = G(x_{\text{init}})$. Further note that the result of Lemma 4.7 is completely independent of the loss function $L_t$, and thus $R$ only depends on the initialization $x_{\text{init}}$ and the parametrization $G$. This finishes the proof. $\qquad\square$

## D.2 Necessary condition for a smooth parametrization to be commuting

*Proof of Theorem 4.9.* Fix any initialization $x_{\text{init}} \in M$, and let the Legendre function $R$ be given such that for all time-dependent loss $L_t$, the gradient flow under $L_t \circ G$ initialized at $x$ can be written as the mirror flow under $L_t$ with respect to the Legendre function $R$. We first introduce a few notations that will be useful for the proof. For any $s \in \mathbb{R}$, we define a time-shifting operator $\mathcal{T}_s$ such that for any time-dependent loss $L_t(\cdot)$, $(\mathcal{T}_s L)_t(\cdot) = L_{t-s}(\cdot)$. We say a time-dependent loss $L_t$ is supported on finite time if $L_t = \sum_{i=1}^k \mathbb{1}_{t \in [t_i, t_{i+1})} L^{(i)}$ for some $k \geq 1$ where $t_1 = 0$, $t_{k+1} = \infty$ and $L^{(k)} \equiv 0$, and we denote $\operatorname{len}(L) = t_k$. We further define the concatenation of two time-dependent loss $L_t, L'_t$ supported on finite time as $L \parallel L' = L + \mathcal{T}_{\operatorname{len}(L)} L'$. We also use $\overline{L}$ to denote the time-reverse of the time-dependent loss $L$ which is supported on finite time, that is, $\overline{L}_t = L_{\operatorname{len}(L)-t}$ for all $t \geq 0$. For any $j \in [d]$ and $\delta > 0$, we define the following loss function

$$\ell_t^{j,\delta}(w) = \mathbb{1}_{0 \leq t \leq \delta} \cdot \langle e_j, w \rangle \tag{15}$$

where $e_j$ is the $j$-th canonical base of $\mathbb{R}^d$.

Now for any $k \geq 2$, let $\{j_i\}_{i=1}^k$ be any sequence where each $j_i \in [d]$. Then we recursively define a sequence of time-dependent losses as follows: First define $L^{1,\delta} = -\ell^{j_1,\delta}$, then sequentially for each $i = 2, 3, \ldots, k$, we define

$$L^{i,\delta} = L^{i-1,\sqrt{\delta}} \parallel \left(-\ell^{j_i,\sqrt{\delta}}\right) \parallel \left(-\overline{L}^{i-1,\sqrt{\delta}}\right) \parallel \ell^{j_i,\sqrt{\delta}} \tag{16}$$

where we write $\overline{L}^{i-1,\sqrt{\delta}} = \overline{L^{i-1,\sqrt{\delta}}}$ for convenience. Denote $\iota_i(\delta) = \operatorname{len}(L^{i,\delta})$ for each $i \in [k]$. Then $\iota_1(\delta) = \delta$ and $\iota_i(\delta) = 2\sqrt{\delta} + 2\iota_{i-1}(\sqrt{\delta})$ for $i = 2, 3, \ldots, k$, which further implies

$$\iota_i(\delta) = \sum_{m=1}^{i-1} 2^m \delta^{1/2^m} + 2^{i-1}\delta^{1/2^{i-1}} \quad \text{for all } i \in [k].$$

Moreover, for each $i = 2, 3, \ldots, k$, the gradient of $L^{i,\delta}$ with respect to $w$ is given by

$$
\nabla L_t^{i,\delta}(w) = \begin{cases}
\nabla L_t^{i-1,\sqrt{\delta}}(w) & 0 \le t \le \iota_{i-1}(\sqrt{\delta}), \\
-e_{j_i} & \iota_{i-1}(\sqrt{\delta}) < t \le \iota_{i-1}(\sqrt{\delta}) + \sqrt{\delta}, \\
-\nabla \overline{L}_t^{i-1,\sqrt{\delta}}(w) & \iota_{i-1}(\sqrt{\delta}) + \sqrt{\delta} < t \le 2\iota_{i-1}(\sqrt{\delta}) + \sqrt{\delta}, \\
e_{j_i} & 2\iota_{i-1}(\sqrt{\delta}) + \sqrt{\delta} < t \le 2\iota_{i-1}(\sqrt{\delta}) + 2\sqrt{\delta}, \\
0 & t > 2\iota_{i-1}(\sqrt{\delta}) + 2\sqrt{\delta}.
\end{cases}
\tag{17}
$$

This inductively implies that for any $t \in [0, \iota_k(\delta)]$, $\nabla L_t^{k,\delta}(w) \in \{e_j\}_{j=1}^d$ does not depend on $w$ and is only determined by $t$. Therefore, for any initialization $x \in M$, for all sufficiently small $\delta > 0$, the gradient flow under $L^{k,\delta}$ for $\iota_k(\delta)$ time, i.e., $\phi_{L^{k,\delta}}^{\iota_k(\delta)}(x)$, is well-defined. Moreover, it follows from (17) that

$$
\int_0^{\iota_{k-1}(\delta)} \nabla L_t^{k,\delta}(w(t)) \mathrm{d}t = \int_0^{\iota_{k-1}(\sqrt{\delta})} \nabla L^{k-1,\sqrt{\delta}}(w(t)) \mathrm{d}t + \int_{\iota_{k-1}(\sqrt{\delta})}^{\iota_{k-1}(\sqrt{\delta})+\sqrt{\delta}} -e_{j_k} \mathrm{d}t
$$

$$
+ \int_{\iota_{k-1}(\sqrt{\delta})+\sqrt{\delta}}^{2\iota_{k-1}(\sqrt{\delta})+\sqrt{\delta}} -\nabla \overline{L}^{k-1,\sqrt{\delta}}(w(t)) \mathrm{d}t + \int_{2\iota_{k-1}(\sqrt{\delta})+\sqrt{\delta}}^{2\iota_{k-1}(\sqrt{\delta})2\sqrt{\delta}} e_{j_k} \mathrm{d}t
$$

$$
= \int_0^{\iota_{k-1}(\sqrt{\delta})} \left( \nabla L_t^{k-1,\sqrt{\delta}}(w(t)) - \nabla \overline{L}_t^{k-1,\sqrt{\delta}}(w(t)) \right) \mathrm{d}t = 0
$$

where the last two equalities follow from the fact that $\nabla L_t^{k-1,\sqrt{\delta}}(w)$ does not depend on $w$ and is only determined by $t$ by our construction.

Hence, the mirror flow with respect to the Legendre function $R$ for the time-dependent loss $L^{k,\delta}$ will return to the initialization after $\iota_k(\delta)$ time since

$$
\nabla R(w(\iota_k(\delta))) - \nabla R(w(0)) = \int_0^{\iota_k(\delta)} -\nabla L^{k,\delta}(w(t)) \mathrm{d}t = 0.
$$

This further implies that

$$
G(x_{\mathrm{init}}) = G\big( \phi_{L^{k,\delta} \circ G}^{\iota_k(\delta)}(x_{\mathrm{init}}) \big)
$$

for all sufficiently small $\delta$. Then differentiating with $\delta$ on both sides yields

$$
\partial G(x) \cdot \left. \frac{\mathrm{d}\phi_{L^{k,\delta} \circ G}^{\iota_k(\delta)}(x_{\mathrm{init}})}{\mathrm{d}\delta} \right|_{\delta=0} = 0.
\tag{18}
$$

Note that if the following holds:

$$
\left. \frac{\mathrm{d}\phi_{L^{k,\delta} \circ G}^{\iota_k(\delta)}(x_{\mathrm{init}})}{\mathrm{d}\delta} \right|_{\delta=0} = [[[[\nabla G_{j_1}, \nabla G_{j_2}], \ldots], \nabla G_{j_{k-1}}], \nabla G_{j_k}](x_{\mathrm{init}}),
\tag{19}
$$

then combining (18) and (19) completes the proof, so it remains to verify (19).

We will prove by induction over $k$, and now let $\{j_i\}_{i=1}^\infty$ be an arbitrary sequence where each $j_i \in [d]$. For notational convenience, we denote for each $k \ge 1$,

$$
\pi_{k,\delta}(\cdot) := \phi_{-\ell^{j_k},\delta}^\delta(\cdot) \quad \text{and} \quad \Pi_{k,\delta}(\cdot) := \phi_{L^{k,\delta}}^{\iota_k(\delta)}(\cdot).
$$

Then their inverse maps are given by $\pi_{k,\delta}^{-1}(\cdot) = \phi_{\ell^{j_k},\delta}^\delta(\cdot)$ and $\Pi_{k,\delta}^{-1}(\cdot) = \phi_{-\overline{L}^{k,\delta}}^{\iota_k(\delta)}(\cdot)$ respectively. Since $G$ is smooth, each $\Pi_{k,\sqrt{\delta}}$ is a $\mathcal{C}^\infty$ function of $\delta^{1/2^k}$, and we can expand it in $\delta^{1/2^k}$ as

$$
\Pi_{k,\sqrt{\delta}}(x) = x + \sum_{i=1}^{2^k} \frac{\delta^{i/2^k}}{i!} \Delta_{k,i}(x) + r_{k,\delta}(x)
\tag{20}
$$

where the remainder term $r_{k,\delta}(x)$ is continuous in $x$ and for each $x \in M$, $r_{k,\delta}(x) = o(\delta)$ (i.e., $\lim_{\delta \to 0} \frac{r_{k,\delta}(x)}{\delta} = 0$), and each $\Delta_{k,i}$ is defined as

$$\Delta_{k,i}(x) = \left. \frac{\mathrm{d}^i \Pi_{k,\sqrt{\delta}}(x)}{\mathrm{d}(\delta^{1/2^k})^i} \right|_{\delta=0}.$$

In particular, for $k = 1$, we have

$$\Pi_{1,\sqrt{\delta}}(x) = \pi_{1,\sqrt{\delta}}(x) = x + \sqrt{\delta} \nabla G_{j_1}(x) + \frac{\delta}{2} \partial(\nabla G_{j_1})(x) \nabla G_{j_1}(x) + r_{1,\delta}(x) \qquad (21)$$

where the second equality holds as well for any other $G_j$ in place of $G_{j_1}$, with a different but similar remainder term. For any fixed $K \geq 2$, there is a small open neighborhood of $x_{\mathrm{init}}$ on $M$, denoted by $\mathcal{N}_{x_{\mathrm{init}}} \subseteq M$, such that for all $k \in [K]$, we have $r_{k,\delta}(x) = o(\delta)$ uniformly over all $x \in \mathcal{N}_{x_{\mathrm{init}}}$, so we can replace all $r_{k,\delta}(x)$ by $o(\delta)$ when $x \in \mathcal{N}_{x_{\mathrm{init}}}$. Then we claim that for each $k = 2, 3, \ldots, K$,

$$\lim_{\delta \to \infty} \frac{1}{\sqrt{\delta}} \sum_{i=1}^{2^{k-1}} \frac{\delta^{i/2^k}}{i!} \Delta_{k,i}(x) = [[[\nabla G_{j_1}, \nabla G_{j_2}], \ldots], \nabla G_{j_k}](x), \quad \forall x \in \mathcal{N}_{x_{\mathrm{init}}}, \qquad (22)$$

which directly implies (19). With a slight abuse of notation, the claim is also true for $k = 1$ since $\Delta_{1,1}(x) = \nabla G_{j_1}(x)$ by (21), so we use this as the base case of the induction. Then, assuming (22) holds for $k - 1 < K$, we proceed to prove it for $k$. For convenience, further define $\mathrm{Lie}_G(j_{1:k}) = [[[\nabla G_{j_1}, \nabla G_{j_2}], \ldots], \nabla G_{j_k}]$.

Combining the Taylor expansion in (20) and (22) for $k - 1$, we obtain for all $x \in \mathcal{N}_{x_{\mathrm{init}}}$ that

$$\Pi_{k-1,\sqrt{\delta}}(x) = x + \sqrt{\delta} \cdot \mathrm{Lie}_G(j_{1:(k-1)})(x) + \sum_{i=2^{k-2}+1}^{2^{k-1}} \frac{\delta^{i/2^{k-1}}}{i!} \Delta_{k-1,i}(x) + o(\delta)$$

for sufficiently small $\delta$. Further apply (21) with $G_{j_k}$ in place of $G_{j_1}$ for sufficiently small $\delta$, and then

$$\Pi_{k-1,\sqrt{\delta}}\big(\pi_{k,\sqrt{\delta}}(x)\big)$$

$$= \Pi_{k-1,\sqrt{\delta}}\left( x + \sqrt{\delta} \nabla G_{j_k}(x) + \frac{\delta}{2} \partial(\nabla G_{j_k})(x) \nabla G_{j_k}(x) + o(\delta) \right)$$

$$= x + \sqrt{\delta} \nabla G_{j_k}(x) + \frac{\delta}{2} \partial(\nabla G_{j_k})(x) \nabla G_{j_k}(x) + o(\delta)$$

$$+ \sqrt{\delta} \cdot \mathrm{Lie}_G(j_{1:(k-1)})\left( x + \sqrt{\delta} \nabla G_{j_k}(x) + \frac{\delta}{2} \partial(\nabla G_{j_k})(x) \nabla G_{j_k}(x) + o(\delta) \right)$$

$$+ \sum_{i=2^{k-2}+1}^{2^{k-1}} \frac{\delta^{i/2^{k-1}}}{i!} \Delta_{k-1,i}\left( x + \sqrt{\delta} \nabla G_{j_k}(x) + \frac{\delta}{2} \partial(\nabla G_{j_k})(x) \nabla G_{j_k}(x) + o(\delta) \right)$$

$$+ r_{k-1,\delta}\left( x + \sqrt{\delta} \nabla G_{j_k}(x) + \frac{\delta}{2} \partial(\nabla G_{j_k})(x) \nabla G_{j_k}(x) + o(\delta) \right)$$

where the second equality follows from the Taylor expansion of $\Pi_{k-1,\sqrt{\delta}}$ and that $\pi_{k,\sqrt{\delta}}(x) \in \mathcal{N}_{x_{\mathrm{init}}}$ for sufficiently small $\delta$. Then by the Taylor expansion of $\mathrm{Lie}_G(j_{1:(k-1)})$ and each $\Delta_{k-1,i}$, we have for all $x \in \mathcal{N}_{x_{\mathrm{init}}}$,

$$\Pi_{k-1,\sqrt{\delta}}\big(\pi_{k,\sqrt{\delta}}(x)\big) = x + \sqrt{\delta} \nabla G_{j_k}(x) + \sqrt{\delta} \cdot \mathrm{Lie}_G(j_{1:(k-1)})(x) + \frac{\delta}{2} \partial(\nabla G_{j_k})(x) \nabla G_{j_k}(x)$$

$$+ \delta \cdot \partial \mathrm{Lie}_G(j_{1:(k-1)})(x) \nabla G_{j_k}(x) + \sum_{i=2^{k-2}+1}^{2^{k-1}} \frac{\delta^{i/2^{k-1}}}{i!} \Delta_{k-1,i}(x) + o(\delta)$$

$$(23)$$

for sufficiently small $\delta$. For the other way around, we similarly have

$$
\begin{aligned}
\pi_{k,\sqrt{\delta}}\big(\Pi_{k-1,\sqrt{\delta}}(x)\big) &= \pi_{k,\sqrt{\delta}}\Bigg(x + \sqrt{\delta}\cdot\mathrm{Lie}_G(j_{1:(k-1)})(x) + \sum_{i=2^{k-2}+1}^{2^{k-1}}\frac{\delta^{i/2^{k-1}}}{i!}\Delta_{k-1,i}(x) + o(\delta)\Bigg) \\
&= x + \sqrt{\delta}\nabla G_{j_k}(x) + \sqrt{\delta}\cdot\mathrm{Lie}_G(j_{1:(k-1)}) + \frac{\delta}{2}\partial(\nabla G_{j_k})(x)\nabla G_{j_k}(x) \\
&\quad + \delta\partial(\nabla G_{j_k})(x)\mathrm{Lie}_G(j_{1:(k-1)})(x) + \sum_{i=2^{k-2}+1}^{2^{k-1}}\frac{\delta^{i/2^k}}{i!}\Delta_{k-1,i}(x) + o(\delta)
\end{aligned}
\tag{24}
$$

for all $x \in \mathcal{N}_{x_{\mathrm{init}}}$, when $\delta$ is sufficiently small. Note that $x = \pi_{k,\sqrt{\delta}}^{-1}\circ\Pi_{k-1,\sqrt{\delta}}^{-1}\circ\Pi_{k-1,\sqrt{\delta}}\circ\pi_{k,\sqrt{\delta}}(x)$, thus

$$
\begin{aligned}
\Pi_{k,\delta}(x) - x &= \pi_{k,\sqrt{\delta}}^{-1}\circ\Pi_{k-1,\sqrt{\delta}}^{-1}\circ\pi_{k,\sqrt{\delta}}\circ\Pi_{k-1,\sqrt{\delta}}(x) - x \\
&= \pi_{k,\sqrt{\delta}}^{-1}\circ\Pi_{k-1,\sqrt{\delta}}^{-1}\circ\pi_{k,\sqrt{\delta}}\circ\Pi_{k-1,\sqrt{\delta}}(x) - \pi_{k,\sqrt{\delta}}^{-1}\circ\Pi_{k,\sqrt{\delta}}^{-1}\circ\Pi_{k,\sqrt{\delta}}\circ\pi_{k,\sqrt{\delta}}(x) \\
&= \pi_{k,\sqrt{\delta}}^{-1}\circ\Pi_{k-1,\sqrt{\delta}}^{-1}\circ\pi_{k,\sqrt{\delta}}\circ\Pi_{k-1,\sqrt{\delta}}(x) - \pi_{k,\sqrt{\delta}}\circ\Pi_{k-1,\sqrt{\delta}}(x) \\
&\quad + \pi_{k,\sqrt{\delta}}\circ\Pi_{k-1,\sqrt{\delta}}(x) - \Pi_{k,\sqrt{\delta}}\circ\pi_{k,\sqrt{\delta}}(x) \\
&\quad + \Pi_{k-1,\sqrt{\delta}}(x)\circ\pi_{k,\sqrt{\delta}} - \pi_{k,\sqrt{\delta}}^{-1}\circ\Pi_{k,\sqrt{\delta}}^{-1}\circ\Pi_{k,\sqrt{\delta}}\circ\pi_{k,\sqrt{\delta}}(x) \\
&= \Pi_{k-1,\sqrt{\delta}}\circ\pi_{k,\sqrt{\delta}}(x) - \pi_{k,\sqrt{\delta}}\circ\Pi_{k-1,\sqrt{\delta}}(x) + o(\delta)
\end{aligned}
\tag{25}
$$

where the last equality follows from the Taylor expansion of $\pi_{k,\sqrt{\delta}}^{-1}\circ\Pi_{k-1,\sqrt{\delta}}^{-1}(\cdot)$ in terms of $\sqrt{\delta}$. Now, combining (23), (24) and (25), we obtain

$$
\begin{aligned}
\Pi_{k,\delta}(x) - x &= \delta\left(\partial(\nabla G_{j_k})(x)\mathrm{Lie}_G(j_{1:(k-1)})(x) - \partial\mathrm{Lie}_G(j_{1:(k-1)})(x)\nabla G_{j_k}(x)\right) + o(\delta) \\
&= \delta\cdot[\mathrm{Lie}_G(j_{1:(k-1)}), \nabla G_{j_k}](x) + o(\delta)
\end{aligned}
\tag{26}
$$

where the second equality follows from the definition of Lie bracket. Comparing (26) with (20) yields (22). This completes the induction for $k \in [K]$ and hence finishes the proof as $K$ is arbitrary. $\square$

*Proof of Corollary 4.11.* It turns out that the necessary condition in Theorem 4.9 is already violated by only considering the Lie algebra spanned by $\{\nabla G_{11}, \nabla G_{12}\}$. We follow the notation in Example 4.4 to define each $E_{ij} \in \mathbb{R}^d$ as the one-hot matrix with the $(i,j)$-th entry being 1, and denote $\overline{E}_{ij} = \frac{1}{2}(E_{ij} + E_{ji})$ and $\Delta_{ij} = E_{ij} - E_{ji}$. Then $[\nabla G_{11}, \nabla G_{12}](U) = 4(\overline{E}_{11}\overline{E}_{12} - \overline{E}_{12}\overline{E}_{11})U = \Delta_{12}U$ and $[\nabla G_{11}, [\nabla G_{11}, \nabla G_{12}]](U) = (\overline{E}_{11}\Delta_{12} - \Delta_{12}\overline{E}_{11})U = \overline{E}_{12}U$. Further noting that $\langle[\nabla G_{11}, [\nabla G_{11}, \nabla G_{12}]], \nabla G_{12}\rangle = 2\left\|\overline{E}_{12}U\right\|_F^2 = \frac{1}{2}\sum_{i=1}^r(U_{1i}^2 + U_{2i}^2)$ must be positive at some $U$ in every open set $M$, by Theorem 4.9, we know such $U_{\mathrm{init}}$ and $L_t$ exist. Moreover, $L_t$ will only depend on $G_{11}(U)$ and $G_{12}(U)$. $\square$

*Proof of Corollary 4.12.* By the condition (b) and Theorem 4.9, we know that each Lie bracket $[\nabla G_i, \nabla G_j] \in \ker(\partial G)$. By the condition (a), we know that each Lie bracket $[\nabla G_i, \nabla G_j] \in \mathrm{span}\{\nabla G_i\}_{i=1}^d$. Combining these two facts, we conclude that each $[\nabla G_i, \nabla G_j] \equiv 0$, so $G$ is a commuting parametrization. $\square$

### D.3 Convergence for gradient flow with commuting parametrization

*Proof of Theorem 4.13.* Recall that the dynamics of $w(t)$ is given by

$$
\mathrm{d}w(t) = -\nabla^2 R(w(t))^{-1}\nabla L(w(t))\mathrm{d}t, \qquad w(0) = G(x_{\mathrm{init}}).
$$

By Lemma 4.7, we know that $R$ is a Legendre function. Therefore, when $R$ is further a Bregman function, we can apply Theorem B.8 to obtain the convergence of $w(t)$. This finishes the proof. $\square$

Based on Theorem B.7, we can prove the trajectory convergence of $w(t)$ for the special case where $\Omega_w(x_{\mathrm{init}}; G) = \mathbb{R}^d$ as summarized in Corollary 4.14.

*Proof of Corollary 4.14.* It suffices to verify that $R$ is a Bregman function in this case. By Lemma 4.7, we know that $R$ is a Legendre function and satisfies that $\mathbb{R}^d = \Omega_w(x_{\mathrm{init}}; G) = \mathrm{dom}\,\nabla R \subseteq \mathrm{dom}\,R \subseteq \mathbb{R}^d$, which implies $\mathrm{dom}\,R = \mathbb{R}^d$. Moreover, the domain of its convex conjugate $Q$ is also $\mathbb{R}^d$. Then by Theorem B.7, we see that $R$ is a Bregman function. This finishes the proof. $\qquad\square$

Next, we prove that for a class of commuting quadratic parametrizations, the corresponding Legendre function is also a Bregman function, thus guaranteeing the trajectory convergence.

*Proof of Theorem 4.15.* Since $A_1, A_2, \ldots, A_d$ commute with each other, these matrices can be simultaneously diagonalized. Thus we can assume without loss of generality that each $A_i = \mathrm{diag}(\lambda_i)$ where $\lambda_i \in \mathbb{R}^D$, then $G_i(x) = \lambda_i^\top x^{\odot 2}$. For convenience, we denote $\Lambda = (\lambda_1, \lambda_2, \ldots, \lambda_d)^\top \in \mathbb{R}^{d \times D}$, so the parametrization is given by $G(x) = \Lambda x^{\odot 2}$. Note that for each $i \in [d]$, $\nabla G_i(x) = 2\lambda_i \odot x$ and $\nabla^2 G_i(x) = 2\mathrm{diag}(\lambda_i)$, so for any $i, j \in [d]$, we have

$$[\nabla G_i, \nabla G_j](x) = 4\mathrm{diag}(\lambda_i)\lambda_j \odot x - 4\mathrm{diag}(\lambda_j)\lambda_i \odot x = 0.$$

Therefore, we see that $G : \mathbb{R}_+^D \to \mathbb{R}^d$ is a commuting parametrization. Also, for any $t \in \mathbb{R}$, $x(t) = x_{\mathrm{init}} - \int_0^t \nabla G_i(x(s))\mathrm{d}s = x_{\mathrm{init}} \odot e^{-2\lambda_i t}$, which proves the first and the second claims. Moreover, if the sign of each coordinate of $x$ will not change from that of initialization, (sign means $+,-$ or $0$). Without loss of generality, below we will assume every coordinate is non-zero at initialization (otherwise we just ignore it). We can also assume the coordinates at initialization are all positive, as the negatives will induce the same trajectory in terms of $G(x)$. By Theorem 4.8, the dynamics of $w(t) = G(x(t))$ is given by

$$\mathrm{d}w(t) = -\nabla^2 R(w(t))^{-1}\nabla L(w(t))\mathrm{d}t, \qquad w(0) = G(x_{\mathrm{init}})$$

for some Legendre function $R$ whose conjugate is denoted by $Q$. To apply the results in Theorem 4.13, it suffices to show that this $R$ is a Bregman function.

To do so, we further denote $\widetilde{w} = x^{\odot 2}$ and $\widetilde{G}(x) = x^{\odot 2}$, then $w = \Lambda \widetilde{w}$ and in this case $\widetilde{G}$ is a commuting parametrization for $\widetilde{w}$ defined on $M = \mathbb{R}_+^D$. Also, we have $\partial G(x) = \Lambda \partial \widetilde{G}(x)$. Let $\widetilde{L} : \mathbb{R}^d \to \mathbb{R}$ be defined by $\widetilde{L}(\widetilde{w}) = L(\Lambda \widetilde{w})$, which satisfies that $\nabla \widetilde{L}(\widetilde{w}) = \Lambda^\top \nabla L(\Lambda \widetilde{w})$. Then the gradient flow with parametrization $\widetilde{G}$ governed by $-\nabla(\widetilde{L} \circ \widetilde{G})(x)$ is given by

$$\begin{aligned}
\mathrm{d}x(t) &= -\nabla(\widetilde{L} \circ \widetilde{G})(x)\mathrm{d}t = -\partial\widetilde{G}(x(t))^\top \nabla\widetilde{L}(\widetilde{G}(x(t))\mathrm{d}t \\
&= -\partial\widetilde{G}(x(t))^\top \Lambda^\top \nabla L(\Lambda\widetilde{G}(x(t))\mathrm{d}t \\
&= -\partial G(x(t))^\top \nabla L(G(x(t))\mathrm{d}t,
\end{aligned}$$

which yields the same dynamics of the gradient flow with parametrization $G$ governed by $-\nabla(L \circ G)(x)$. Therefore, we have $w(t) = G(x(t)) = \Lambda\widetilde{G}(x(t)) = \Lambda\widetilde{w}(t)$, where again by Theorem 4.8, the dynamics of $\widetilde{w}(t)$ is

$$\mathrm{d}\widetilde{w}(t) = -\nabla^2\widetilde{R}(\widetilde{w}(t))^{-1}\nabla\widetilde{L}(\widetilde{w}(t))\mathrm{d}t, \qquad \widetilde{w}(0) = \widetilde{G}(x_{\mathrm{init}})$$

for some Legendre function $\widetilde{R}$ whose conjugate is denoted by $\widetilde{Q}$. For any $x \in M$ and $\widetilde{\mu} \in \mathbb{R}^D$, we define $\widetilde{\psi}(x; \widetilde{\mu}) = \phi_{\widetilde{G}_1}^{\widetilde{\mu}_1} \circ \phi_{\widetilde{G}_2}^{\widetilde{\mu}_2} \circ \cdots \circ \phi_{\widetilde{G}_D}^{\widetilde{\mu}_D}(x)$. We need the following lemma.

**Lemma D.3.** *In the setting of the proof of Theorem 4.15, for any $\mu \in \mathbb{R}^d$ and $x \in M$, we have $\psi(x; \mu) = \widetilde{\psi}(x; \Lambda^\top \mu)$.*

Recall from Lemma 4.7 that $\nabla Q(\mu) = G(\psi(x_{\mathrm{init}}; \mu))$ for any $\mu \in \mathbb{R}^d$ and $\nabla\widetilde{Q}(\widetilde{\mu}) = \widetilde{G}(\widetilde{\psi}(x_{\mathrm{init}}; \widetilde{\mu}))$ for any $\widetilde{\mu} \in \mathbb{R}^D$. Note that

$$\nabla Q(\mu) = \Lambda\psi(x_{\mathrm{init}}; \mu)^{\odot 2} = \Lambda\widetilde{\psi}(x_{\mathrm{init}}; \Lambda^\top\mu)^{\odot 2} = \Lambda\widetilde{G}(\widetilde{\psi}(x_{\mathrm{init}}; \Lambda^\top\mu)) = \Lambda\nabla\widetilde{Q}(\Lambda^\top\mu) \quad (27)$$

where the second equality follows from Lemma D.3. This implies that $Q(\mu) = \widetilde{Q}(\Lambda^\top\mu) + C$ for some constant $C$. Recall the definition of convex conjugate, and we have

$$\widetilde{R}(\widetilde{w}) = \sup_{\widetilde{\mu}\in\mathbb{R}^D} \langle\widetilde{\mu}, \widetilde{w}\rangle - \widetilde{Q}(\widetilde{\mu}), \qquad R(w) = \sup_{\mu\in\mathbb{R}^d} \langle\mu, w\rangle - Q(\mu).$$

Then for any $\widetilde{w} \in \mathbb{R}^D$, we have

$$R(\Lambda \widetilde{w}) = \sup_{\mu \in \mathbb{R}^d} \langle \mu, \Lambda \widetilde{w} \rangle - Q(\mu) = \sup_{\mu \in \mathbb{R}^d} \langle \Lambda^\top \mu, \widetilde{w} \rangle - \widetilde{Q}(\Lambda^\top \mu) - C$$

$$= \sup_{\widetilde{\mu} \in \Lambda^\top \mathbb{R}^d} \langle \widetilde{\mu}, \widetilde{w} \rangle - \widetilde{Q}(\widetilde{\mu}) - C \leq \sup_{\widetilde{\mu} \in \mathbb{R}^D} \langle \widetilde{\mu}, \widetilde{w} \rangle - \widetilde{Q}(\widetilde{\mu}) - C = \widetilde{R}(\widetilde{w}) - C \qquad (28)$$

Therefore, for any $\widetilde{w} \in \operatorname{dom} \widetilde{R}$, it holds that $R(\Lambda \widetilde{w}) \leq \widetilde{R}(\widetilde{w}) - C < \infty$, so $\Lambda \operatorname{dom} \widetilde{R} \subseteq \operatorname{dom} R$, where $\Lambda \operatorname{dom} \widetilde{R}$ On the other hand, by (27) and Proposition B.3, we have

$$\operatorname{dom} \nabla R = \operatorname{range} \nabla Q \subseteq \Lambda \operatorname{range} \nabla \widetilde{Q} = \Lambda \operatorname{dom} \nabla \widetilde{R}$$

and it follows that

$$\operatorname{int}(\operatorname{dom} R) = \operatorname{dom} \nabla R \subseteq \Lambda \operatorname{dom} \nabla \widetilde{R} = \Lambda \operatorname{int}(\operatorname{dom} \widetilde{R}).$$

Combining the above, we see that $\operatorname{dom} R = \Lambda \operatorname{dom} \widetilde{R}$. As discussed in Section 1, here it is straightforward to verify that $\widetilde{R}(\widetilde{w}) = \sum_{i=1}^D \widetilde{w}_i (\ln \frac{\widetilde{w}_i}{x_{\mathrm{init},i}^2} - 1)$, which is indeed a Bregman function with domain $\operatorname{dom} \widetilde{R} = \overline{\mathbb{R}_+^D}$. Thus $\operatorname{dom} R = \Lambda \overline{\mathbb{R}_+^D}$ is also a closed set. This yields the first condition in Definition B.6.

Next, we verify the second condition in Definition B.6. For any $\mu \in \mathbb{R}^d$, we have

$$\nabla R(G(\psi(x_{\mathrm{init}}; \mu))) = \nabla R(\nabla Q(\mu)) = \mu$$

and

$$\nabla \widetilde{R}(\widetilde{G}(\psi(x_{\mathrm{init}}; \mu))) = \nabla \widetilde{R}(\widetilde{G}(\widetilde{\psi}(x_{\mathrm{init}}; \Lambda^\top \mu))) = \nabla \widetilde{R}(\nabla \widetilde{Q}(\Lambda^\top \mu)) = \Lambda^\top \mu.$$

Comparing the above two equalities, we get

$$\nabla \widetilde{R}(\widetilde{w}) = \Lambda^\top \nabla R(\Lambda \widetilde{w}) \qquad (29)$$

for all $\widetilde{w} \in \mathbb{R}_+^D$. Then for any $\widetilde{w} \in \overline{\mathbb{R}_+^D}$ and $y = \Lambda \widetilde{y} \in \operatorname{int}(\operatorname{dom} R)$, we have

$$\begin{aligned} D_R(\Lambda \widetilde{w}, y) &= R(\Lambda \widetilde{w}) - R(y) - \langle \nabla R(y), \Lambda \widetilde{w} - y \rangle \\ &= R(\Lambda \widetilde{w}) - R(\Lambda \widetilde{y}) - \langle \Lambda^\top \nabla R(\Lambda \widetilde{y}), \widetilde{w} - \widetilde{y} \rangle \\ &= R(\Lambda \widetilde{w}) - R(\Lambda \widetilde{y}) - \langle \nabla \widetilde{R}(\widetilde{y}), \widetilde{w} - \widetilde{y} \rangle \\ &= R(\Lambda \widetilde{w}) - R(\Lambda \widetilde{y}) - \widetilde{R}(\widetilde{w}) + \widetilde{R}(\widetilde{y}) + D_{\widetilde{R}}(\widetilde{w}, \widetilde{y}) \qquad (30) \\ &\geq R(\Lambda \widetilde{w}) - \widetilde{R}(\widetilde{w}) + C + D_{\widetilde{R}}(\widetilde{w}, \widetilde{y}) \end{aligned}$$

where the inequality follows from (28). Therefore, we further have for any $\alpha \in \mathbb{R}$

$$\{y \in \operatorname{int}(\operatorname{dom} R) \mid D_R(\Lambda \widetilde{w}, y) \leq \alpha\} \subseteq \Lambda \{\widetilde{y} \in \mathbb{R}_+^D \mid D_{\widetilde{R}}(\widetilde{w}, \widetilde{y}) \leq \alpha - R(\Lambda \widetilde{w}) + \widetilde{R}(\widetilde{w}) - C\}$$

where the right-hand side is bounded since $\widetilde{R}$ is a Bregman function, and so is the left-hand side.

Finally, we verify the third condition in Definition B.6. Consider any $w \in \operatorname{dom} R$ and sequence $\{w_i\}_{i=1}^\infty \subset \operatorname{int}(\operatorname{dom} R)$ such that $\lim_{i \to \infty} w_i = w$. Since $\operatorname{dom} R = \Lambda \operatorname{dom} \widetilde{R}$, there is some $\widetilde{w} \in \overline{\mathbb{R}_+^D}$ such that $w = \Lambda \widetilde{w}$ and some $\widetilde{w}_i \in \mathbb{R}_+^D$ for each $i \in \mathbb{N}^+$ such that $w_i = \Lambda \widetilde{w}_i$. We have that

$$\begin{aligned} R(w) - R(w_i) &= \int_0^1 \langle \nabla R((1-t)w_i + tw), w - w_i \rangle \mathrm{d}t \\ &= \int_0^1 \langle \Lambda^\top \nabla R(\Lambda((1-t)\widetilde{w}_i + t\widetilde{w})), \widetilde{w} - \widetilde{w}_i \rangle \mathrm{d}t \\ &= \int_0^1 \langle \nabla \widetilde{R}((1-t)\widetilde{w}_i + t\widetilde{w}), \widetilde{w} - \widetilde{w}_i \rangle \mathrm{d}t \\ &= \widetilde{R}(\widetilde{w}) - \widetilde{R}(\widetilde{w}_i). \end{aligned}$$

Combining this with (30), we get $D_R(w, w_i) = D_{\widetilde{R}}(\widetilde{w}, \widetilde{w}_i)$. Note that we can always choose each $\widetilde{w}_i$ properly such that $\lim_{i \to \infty} \widetilde{w}_i = \widetilde{w}$. Then since $\widetilde{R}$ is a Bregman function, we have

$$\lim_{i \to \infty} D_R(w, w_i) = \lim_{i \to \infty} D_{\widetilde{R}}(\widetilde{w}, \widetilde{w}_i) = 0.$$

Therefore, we conclude that $R$ is also a Bregman function. This finishes the proof. $\qquad \square$

*Proof of Lemma D.3.* For each $i \in [D]$ and any $t > 0$, we have

$$\phi_{G_i}^t(x) = x + \int_{s=0}^{t} -\nabla G_i(\phi_{f_i}^s(x)) \mathrm{d}s = x + \int_{s=0}^{t} -\sum_{j=1}^{D} \lambda_{i,j} \nabla \widetilde{G}_j(\phi_{f_i}^s(x)) \mathrm{d}s = \widetilde{\psi}(x; t\lambda_i)$$

where the last equality follows from Lemma 4.6. Therefore, for any $\mu \in \mathbb{R}^d$, we further have

$$\begin{aligned}
\psi(x; \mu) &= \phi_{G_1}^{\mu_1} \circ \phi_{G_2}^{\mu_2} \circ \cdots \circ \phi_{G_d}^{\mu_d}(x) \\
&= \phi_{\widetilde{G}_1}^{\mu_1 \lambda_{1,1}} \circ \cdots \circ \phi_{\widetilde{G}_D}^{\mu_1 \lambda_{1,D}} \circ \cdots \circ \phi_{\widetilde{G}_1}^{\mu_d \lambda_{d,1}} \circ \cdots \circ \phi_{\widetilde{G}_D}^{\mu_d \lambda_{d,D}}(x) \\
&= \phi_{\widetilde{G}_1}^{\sum_{i=1}^{d} \mu_i \lambda_{i,1}} \circ \cdots \circ \phi_{\widetilde{G}_D}^{\sum_{i=1}^{d} \mu_i \lambda_{i,D}}(x) \\
&= \phi_{\widetilde{G}_1}^{(\Lambda^\top \mu)_1} \circ \cdots \circ \phi_{\widetilde{G}_D}^{(\Lambda^\top \mu)_D}(x) = \widetilde{\psi}(x; \Lambda^\top \mu).
\end{aligned}$$

where the third equality follows from the assumption that $\widetilde{G}$ is a commuting parametrization. This finishes the proof. $\square$

## D.4 Results for underdetermined linear regression

Here we provide the proof for the implicit bias result for the quadratically overparametrized linear model.

*Proof of Theorem 4.16.* By Theorem 4.8, $w(t)$ obeys the following mirror flow:

$$\mathrm{d}\nabla R(w(t)) = -\nabla L(w(t))\mathrm{d}t, \qquad w(0) = G(x_{\mathrm{init}}).$$

Applying Theorem 3.8 yields

$$D_R(w_\infty, G(x_{\mathrm{init}})) = \min_{w:Zw=Y} D_R(w, G(x_{\mathrm{init}})).$$

Therefore, for any $w \in \mathrm{int}(\mathrm{dom}\, R)$ such that $Zw = Y$, we have

$$\begin{aligned}
R(w_\infty) &- R(G(x_{\mathrm{init}})) - \langle \nabla R(G(x_{\mathrm{init}})), w_\infty - G(x_{\mathrm{init}}) \rangle \\
&\le R(w) - R(G(x_{\mathrm{init}})) - \langle \nabla R(G(x_{\mathrm{init}})), w - G(x_{\mathrm{init}}) \rangle
\end{aligned}$$

which can be reorganized as

$$R(w_\infty) \le R(w) - \langle \nabla R(G(x_{\mathrm{init}})), w - w_\infty \rangle. \tag{31}$$

Note that by Lemma 4.7, we also have

$$\nabla R(G(x_{\mathrm{init}})) = \nabla R(G(\psi(x_{\mathrm{init}}; 0))) = \nabla R(\nabla Q(0)) = 0 \tag{32}$$

where the last equality follows from the property of convex conjugate. Combining (31) and (32), we get $R(w_\infty) \le R(w)$ for all $w \in \mathrm{int}(\mathrm{dom}\, R)$ such that $Zw = Y$. By the continuity of $R$, this property can be further extended to the entire $\mathrm{dom}\, R$, and for any $w \notin \mathrm{dom}\, R$, we have $R(w) = \infty$ by definition, so $R(w_\infty) \le R(w)$ holds trivially. This finishes the proof. $\square$

*Proof of Corollary 4.17.* By symmetry, we assume without loss of generality that all coordinates of $x_{\mathrm{init}}$ are positive. Note that for $M = \mathbb{R}_+^D$ with $D = 2d$, $G : M \to \mathbb{R}^d$ can be written as $G_i(x) = x^\top A_i x$ where each $A_i = e_i e_i^\top - e_{d+i} e_{d+i}^\top$. Therefore, this parametrization $G$ satisfies the conditions in Theorem 4.15, which then implies the convergence of $w(t)$.

Next, we identify the function $R$ given by Theorem 4.8. we have $\psi(x_{\mathrm{init}}; \mu) = \begin{pmatrix} u_0 \odot e^{-2\mu} \\ v_0 \odot e^{2\mu} \end{pmatrix}$ and thus

$$\begin{aligned}
G(\psi(x_{\mathrm{init}}; \mu)) &= u_0^{\odot 2} \odot e^{-4\mu} - v_0^{\odot 2} \odot e^{4\mu} \\
&= (u_0^{\odot 2} + v_0^{\odot 2}) \odot \sinh(4\mu) + (u_0^{\odot 2} - v_0^{\odot 2}) \odot \cosh(4\mu).
\end{aligned}$$

So $G(\psi(x_{\mathrm{init}}; \mu))$ is the gradient of $Q(\mu) = \frac{1}{4}(u_0^{\odot 2} + v_0^{\odot 2}) \odot \cosh(4\mu) + \frac{1}{4}(u_0^{\odot 2} - v_0^{\odot 2}) \odot \sinh(4\mu) + C$ where $C$ is an arbitrary constant. Also note that $(\nabla Q(\mu))_i$ only depends on $\mu_i$, then we have

$$\begin{aligned}
(\nabla R(w))_i = (\nabla Q(\mu))_i^{-1}(w) &= \frac{1}{4} \ln \left( \sqrt{1 + \left( \frac{w_i}{2u_{0,i} v_{0,i}} \right)^2} + \frac{w_i}{2u_{0,i} v_{0,i}} \right) + \frac{1}{4} \ln \frac{v_{0,i}}{u_{0,i}} \\
&= \frac{1}{4} \operatorname{arcsinh} \left( \frac{w_i}{2u_{0,i} v_{0,i}} \right) + \frac{1}{4} \ln \frac{v_{0,i}}{u_{0,i}}
\end{aligned}$$

which further implies that

$$R(w) = \frac{1}{4} \sum_{i=1}^{d} \left( w_i \operatorname{arcsinh} \left( \frac{w_i}{2u_{0,i} v_{0,i}} \right) - \sqrt{w_i^2 + 4u_{0,i}^2 v_{0,i}^2} - w_i \ln \frac{u_{0,i}}{v_{0,i}} \right) + C.$$

This finishes the proof. □

# E   Omitted proofs in Section 5

We first prove the following intermediate result that will be useful in the proof of Theorem 5.1.

**Lemma E.1.** *Under the setting of Theorem 5.1, let $F$ be the smooth map that isometrically embeds $(\operatorname{int}(\operatorname{dom} R), g^R)$ into $(\mathbb{R}^D, \bar{g})$. Let $M = \operatorname{range}(F)$, and denote the inverse of $F$ by $\widetilde{G} : M \to \mathbb{R}^d$. Then for any $w \in \operatorname{int}(\operatorname{dom} R)$, it holds that*

$$\partial F(w)(\partial F(w)^\top \partial F(w))^{-1} = \partial \widetilde{G}(F(w))^\top \quad and \quad \partial \widetilde{G}(F(w)) \partial \widetilde{G}(F(w))^\top = \nabla^2 R(w)^{-1}.$$

*Proof of Lemma E.1.* For any $x \in M$ and $v \in T_x(M)$, consider a parametrized curve $\{x(t)\}_{t\geq 0} \subset M$ such that $x(0) = x$ and $\frac{\mathrm{d}x(t)}{\mathrm{d}t}\big|_{t=0} = v$. Since $x(t) = F(\widetilde{G}(x(t)))$ for any $t \geq 0$, differentiating with respect to $t$ on both sides and evaluating at $t = 0$ yield

$$v = \partial F(\widetilde{G}(x)) \partial \widetilde{G}(x) v. \tag{33}$$

Now, for any $w \in \operatorname{int}(\operatorname{dom} R)$, let $x = F(w)$, then for any $v \in T_x(M)$, it follows from (33) that

$$v^\top \partial F(w) = v^\top (\partial F(w) \partial \widetilde{G}(F(w)))^\top \partial F(w) = v^\top \partial \widetilde{G}(F(w))^\top \partial F(w)^\top \partial F(w).$$

Note that the span of the column space of $\partial F(w)$ is exactly $T_x(M)$, so for any $v$ in the orthogonal complement of $T_x(M)$, it holds that

$$v^\top \partial F(w) = 0 = v^\top \partial \widetilde{G}(F(w))^\top \partial F(w)^\top \partial F(w)$$

where the second equality follows from the fact that for any $i \in [d]$, $\nabla \widetilde{G}_i(x) \in T_x(M)$. Therefore, combining the above two cases, we conclude that

$$\partial F(w) = \partial \widetilde{G}(F(w))^\top \partial F(w)^\top \partial F(w).$$

Since $\partial F(w)^\top \partial F(w) = \nabla^2 R(w)$ is invertible, we then get

$$\partial \widetilde{G}(F(w))^\top = \partial F(w)(\partial F(w)^\top \partial F(w))^{-1}.$$

Next, for any $w \in \operatorname{int}(\operatorname{dom} R)$, since $\widetilde{G}(F(w)) = w$, differentiating on both sides yields

$$\partial \widetilde{G}(F(w)) \partial F(w) = I_d.$$

Therefore, using the identity proved above, we have

$$\begin{aligned}\partial \widetilde{G}(F(w)) \partial \widetilde{G}(F(w))^\top &= \partial \widetilde{G}(F(w)) \partial F(w)(\partial F(w)^\top \partial F(w))^{-1} \\ &= (\partial F(w)^\top \partial F(w))^{-1} = \nabla^2 R(w)^{-1}.\end{aligned}$$

This finishes the proof. □

*Proof of Theorem 5.1.* By Nash's embedding theorem, there is a smooth map $F : \operatorname{int}(\operatorname{dom} R) \to \mathbb{R}^D$ that isometrically embeds $(\operatorname{int}(\operatorname{dom} R), g^R)$ into $(\mathbb{R}^D, \bar{g})$. Denote $M = \operatorname{range}(F)$, *i.e.*, the embedding of $\operatorname{int}(\operatorname{dom} R)$ in $\mathbb{R}^D$. We further denote the inverse of $F$ on $M$ by $\widetilde{G} : M \to \mathbb{R}^d$. Note $(M, \widetilde{G})$ is a global atlas for $M$, we have that $T_x(M) = \operatorname{span}(\{\nabla \widetilde{G}_i(x)\}_{i=1}^d)$ for all $x \in M$. This $\widetilde{G}$ is almost the commuting parametrization that we seek for, except now it is only defined on $M$ but not on an open neighborhood of $M$. Yet we can extend $\widetilde{G}$ to an open neighbourhood of $M$ in the following way: First by [21], for each $x \in M$, there is an open neighbourhood $U_x$ of $x$ such that projection function $P$ defined by

$$P(y) = \operatorname*{argmin}_{y' \in M} \|y - y'\|_2$$

is smooth in $U_x$. Then we define $U = \cup_{x\in M} U_x$, and extend $\widetilde{G}$ to $U$ by defining $G(x) := \widetilde{G}(P(x))$ for all $x \in U$. We have $G(x) = \widetilde{G}(x)$ for all $x \in M$, and we can verify that $\partial G \equiv \partial\widetilde{G}$ on $M$ as well. For any $v \in T_x(M)$, let $\{\gamma(t)\}_{t\geq 0}$ be a parametrized curve on $M$ such that $\gamma(0) = x$ and $\frac{d\gamma(t)}{dt}\big|_{t=0} = v$, then for sufficiently small $t$, by Taylor expansion we have

$$\gamma(t) = P(\gamma(t)) = P(x) + \partial P(x)(\gamma(t) - x) + o(\|\gamma(t) - x\|_2)$$
$$= x + \partial P(x)(\gamma(t) - x) + o(\|\gamma(t) - x\|_2)$$

which implies that $v = \partial P(x)v$ by letting $t \to 0$. While for any $v$ in the orthogonal complement of $T_x(M)$, for sufficiently small $\delta > 0$, we have $P(x + \delta v)$ is smooth in $\delta$. Then since $P(x + \delta v) \in M$ for all sufficiently small $\delta$ by its definition, we have

$$\partial P(x)v = \frac{dP(x + \delta v)}{d\delta}\bigg|_{\delta=0} = \lim_{\delta\to 0} \frac{P(x + \delta v) - P(x)}{\delta} =: u \in T_x(M). \tag{34}$$

Note that $\|x + \delta v - P(x + \delta v)\|_2 \leq \|x + \delta v - P(x)\|_2 = \delta\|v\|_2$, and by Taylor expansion, we have

$$\|x + \delta v - P(x + \delta v)\|_2 = \|x + \delta v - \delta\partial P(x)v + O(\delta^2)\|_2 = \|x + \delta v - \delta u + O(\delta^2)\|_2$$

where $O(\delta^2)$ denotes a term whose norm is bounded by $C\delta^2$ for a constant $C > 0$ for all sufficiently small $\delta$, and the second equality follows from (34). Then dividing both sides by $\delta$ and letting $\delta \to 0$, we have $\|v\|_2 \geq \|v - u\|_2$. Since $u$ is orthogonal to $v$, we must have $u = 0$. As $v$ is arbitrary, we conclude that $\partial P(x)$ is the orthogonal projection matrix onto $T_x(M)$. Then differentiating both sides of $G(x) = \widetilde{G}(P(x))$ with $x$ yields

$$\partial G(x) = \partial\widetilde{G}(P(x))\partial P(x) = \partial\widetilde{G}(x) \tag{35}$$

where the second equality follows from the fact that $T_x(M) = \mathrm{span}(\{\nabla\widetilde{G}_i(x)\}_{i=1}^d)$. This further implies that the solution of Equation (11) satisfies $dx/dt = -\nabla(L \circ \widetilde{G})(x) \in T_x(M)$, and thus $x(t) \in M$ for all $t \geq 0$.

Now we consider the mirror flow

$$dw(t) = -\nabla^2 R(w(t))^{-1}\nabla L_t(w(t))dt, \qquad w(0) = w_{\mathrm{init}}.$$

Since $\nabla^2 R(w) = \partial F(w)^\top \partial F(w)$ by the fact that $F$ is an isometric embedding, we further have

$$dw(t) = -\big(\partial F(w(t))^\top \partial F(w(t))\big)^{-1}\nabla L_t(w(t))dt.$$

Now define $x(t) = F(w(t))$, and it follows that

$$dx(t) = \partial F(w(t))dw(t) = -\partial F(w(t))(\partial F(w(t))^\top \partial F(w(t)))^{-1}\nabla L_t(w(t))dt$$
$$= -\partial G(F(w(t)))^\top \nabla L_t(w(t))dt = -\nabla(L_t \circ G)(x(t))dt$$

where the third equality follows from Lemma E.1 and (35).

Next, we verify that $G$ restricted on $M$, $\widetilde{G}$, is a commuting and regular parametrization. First, for any $x \in M$, we have $\partial\widetilde{G}(x)^\top = \partial F(\widetilde{G}(x))(\partial F(\widetilde{G}(x))^\top \partial F(\widetilde{G}(x)))^{-1}$ by Lemma E.1 and (35). Since $\nabla^2 R(w) = \partial F(w)^\top \partial F(w)$ is of rank $d$ for all $w \in \mathrm{int}(\mathrm{dom}\, R)$, it follows that $\partial F(w)$ is also of rank $d$ for all $w \in \mathrm{int}(\mathrm{dom}\, R)$, thus $\partial\widetilde{G}(x)$ is of rank $d$ for all $x \in M$. The commutability of $\{\nabla\widetilde{G}_i\}_{i=1}^d$ follows directly from Corollary 4.12. Here we just need to show $\mathrm{rank}(\Omega_x(x; \widetilde{G})) = \mathrm{rank}(M)$. This is because on one hand $\mathrm{rank}(\Omega_x(x; \widetilde{G})) \geq \mathrm{rank}(\mathrm{span}(\{\nabla\widetilde{G}_i(x)\}_{i=1}^d)) = \mathrm{rank}(M)$, and on the other hand, $\mathrm{rank}(\Omega_x(x; \widetilde{G})) \leq \mathrm{rank}(M)$ since $\Omega_x(x; \widetilde{G}) \subset M$, for any $x \in M$.

Finally, we show that when $R$ is a mirror map, each $\nabla\widetilde{G}_j$ is a complete vector field on $M$. For any $x_{\mathrm{init}} \in M$, consider loss $L_t(w) = \langle e_j, w\rangle$, and the corresponding gradient flow is

$$dx(t) = -\nabla(L_t \circ \widetilde{G})(x(t))dt = -\partial\widetilde{G}(x(t))^\top \nabla L_t(\widetilde{G}(x(t)))dt = -\nabla\widetilde{G}_j(x(t)),$$

so $x(t) = \phi_{\widetilde{G}_j}^t(x_{\mathrm{init}})$ for all $t \geq 0$. On the other hand, $w(t) = \widetilde{G}(x(t))$ satisfies that

$$dw(t) = \partial\widetilde{G}(x(t))dx(t) = -\partial\widetilde{G}(x(t))\partial\widetilde{G}(x(t))^\top \nabla L_t(w(t))dt$$
$$= -\nabla^2 R(w(t))^{-1}\nabla L_t(w(t))dt = -\nabla^2 R(w(t))^{-1}e_j dt$$

where the third equality follows from Lemma E.1 and Equation (35). Therefore, rewriting the above as a mirror Flow yields

$$\mathrm{d}\nabla R(w(t)) = -e_j \mathrm{d}t,$$

the solution to which exists for all $t \in \mathbb{R}$ and is given by $\nabla R(w(t)) = e_j t$, so $w(t) = (\nabla R)^{-1}(e_j t)$ is defined for all $t \in \mathbb{R}$ as $\nabla R$ is surjective. This further implies that $x(t) = F(w(t))$ is well-defined for all $t \in \mathbb{R}$, hence $\nabla \widetilde{G}_j$ is a complete vector field. $\qquad \square$