# OpenReview forum: "Implicit Bias of Gradient Descent on Reparametrized Models: On Equivalence to Mirror Descent"
_NeurIPS.cc/2022/Conference — NeurIPS 2022 Accept_

### Official Review · Reviewer_puj1 · 2022-07-12

**Rating:** 6
**Confidence:** 3
**Soundness:** 3 good
**Presentation:** 3 good
**Contribution:** 2 fair

**Summary:**

This paper focuses on implicit bias of gradient descent for reparametrized models.
In this setting, a loss L over parameters w (eg weights of a neural network) is minimized, where these parameters w write w=G(x).
Hence, they consider the gradient flow of L in the x space. In the weight space, the gradient flow writes with a preconditioner (partial G(x) partial G(x))^T.
It is known that if the latter preconditioner is equal to a Hessian map nabla^2 R(w)^-1 where R is strictly convex, then the latter flow is a Riemannian gradient flow (RGF) with respect to this metric tensor nabla^2 R.

In this paper, the authors tackle the question of characterizing the gradient flow , i.e. when given a parametrization G, we can consider the assocciated
RGF with mirror map R; and conversely, for what R does one have a corresponding parametrization G.

The main class at study are commutative parametrizations, that holds for instance when each coordinate of G depends on different subsets of coordinates (separable case). Th 4.4 is the first main result of the paper and states that if G is commutative, then there exists a mirror map R (depending on xinit and G) such that the dynamics of w is a mirror flow. The proof relies at some point on some sufficient condition (given in Lemma C.1, already highlighted in [42]) under which a vector field is the gradient of a convex function. Convergence of the flow is proved if G is a linear reparametrization (G(x)=xAx) in Corollary 4.10. To characterize implicit bias of gradient descent, the authors use Th 3.6 which characterizes implicit bias (ie properties of the solution) of a mirror flow.
Finally the authors study the reciprocal question, i.e. for a mirror map R, does there exist a corresponding commutative reparametrization G, and provide a positive answer in Theorem 5.3 that holds generally.


**Questions:**

- to what extent the models at study (ie commutative parametrizations) are restrictive. Could the authors clarify this point and the limitations of this hypothesis ? how large is the class of models considered?  is it limited to diagonal neural nets? are more complicated architectures ruled out because of the negative result of Remark 4.7?
- th 4.4: intuitively, why the mirror map nabla R should depend on xinit?
- th 4.11: while the result is interesting, since it says that the solution is a minimizer of the convex regularizer R, this function R remains implicit/is not known if I understood well ? in corollary 12 it is explicit.
- l295: the proposed formula for nabla^2 R (with the inverse of G = F) look quite different from the one in Lemma 4.6 involving psi (hence gradient flows of G ), could the authors clarify the link between the mirror map in section 4 vs section 5?




**Limitations:**

Limitations are too briefly evoked (e.g. in the conclusion) but they should be discussed more extensively in my opinion.

**Strengths And Weaknesses:**

Strengths

The paper brings interesting contributions to the growing literature on implicit bias of optimization algorithms, an important problem in the machine learning literature nowadays. I think the characterization of implicit bias of gradient descent through the mirror map R in the reparametrization setting at study, explicitly when it is more explicit in Th 4.11 and Cor 4.12, is novel and interesting.
The paper is quite well written and the results appear technically sound (I read the proofs in the main, but I did not read the proofs in detail in the appendix).

Weaknesses: I think that the paper lacks clarity in its current state. It would be better with more interpretation and discussion of the results, see questions below. Moreover, due to the limitations of the reparametrization model at study and the lack of empirical support, the impact of the paper may be limited.

---

> ### Author Response · Authors · 2022-08-02
> **Response to Reviewer puj1**
>
> We would like to thank the reviewer for acknowledging the contributions of our paper and also for the valuable feedback. Please see our responses to the questions below.
>
> **Q1**: To what extent the models at study (i.e commutative parametrizations) are restrictive. Could the authors clarify this point and the limitations of this hypothesis? how large is the class of models considered? is it limited to diagonal neural nets? are more complicated architectures ruled out because of the negative result of Remark 4.7?
>
> **A1**: We provide a new necessary condition in the appendix of revision and using that we formally exclude the possibility to write gradient flow under matrix factorization ($W=UV^\top$) as a mirror flow.
>
> ---
>
> **Q2**: th 4.4: intuitively, why the mirror map nabla R should depend on $x_{init}$?
>
> **A2**: This is because each $x_{init}$ decides a different orbit (or reachable set) for all gradient flow with respect to different time-dependent loss. Different orbits can have different geometry, i.e. metric tensor. This dependence can also be seen in previous works, e.g., [Woodworth et al., 2020, Azulay et al., 2021].
>
> _Reference_:
>
> - Blake Woodworth, Suriya Gunasekar, Jason D Lee, Edward Moroshko, Pedro Savarese, Itay Golan, Daniel Soudry, and Nathan Srebro. Kernel and rich regimes in overparametrized models. In Conference on Learning Theory, pages 3635–3673. PMLR, 2020
> - Shahar Azulay, Edward Moroshko, Mor Shpigel Nacson, Blake E Woodworth, Nathan Srebro, Amir Globerson, and Daniel Soudry. On the implicit bias of initialization shape: Beyond infinitesimal mirror descent. In International Conference on Machine Learning, pages 468–477. PMLR, 2021.
>
> ---
>
> **Q3**: th 4.11: while the result is interesting, since it says that the solution is a minimizer of the convex regularizer $R$, this function $R$ remains implicit/is not known if I understood well? in corollary 12 it is explicit.
>
> **A3**: The convex regularizer $R$ in Theorem 4.11 is exactly the $R$ given by Lemma 4.6. We know the closed-form expression of $R$ in some cases such as Corollary 12. We will clarify this in the revision.
>
> ---
>
> **Q4**: l295: the proposed formula for $\nabla^2 R$ (with the inverse of $G$ = $F$) look quite different from the one in Lemma 4.6 involving $\psi$ (hence gradient flows of $G$), could the authors clarify the link between the mirror map in section 4 vs section 5?
>
> **A4**: We clarify this as follows: In Lemma 4.6, $\psi(x_{init}; \mu)$ is a point in the reachable set. Then by denoting $x = \psi(x_{init}; \mu)$, the proposed formula for $\nabla^2 R$ writes $\nabla^2 R(G(x)) = (\partial G(x) \partial G(x))^{-1}$, which exactly coincides with the one in line 295 by letting $x = F(w)$.

---

### Official Review · Reviewer_eah9 · 2022-07-14

**Rating:** 8
**Confidence:** 4
**Soundness:** 4 excellent
**Presentation:** 4 excellent
**Contribution:** 3 good

**Summary:**

For a reparametrisation $w = G(x)$ where $x_t$ follows a gradient flow on some loss $L \odot G$, the authors give a sufficient condition on the parametrisation (i.e. commuting parametrisation) so that $w_t$ to corresponds a mirror flow on $L$ and with a related mirror map.  They then show that any mirror flow can be seen as the commutative reparametrisation of a gradient flow.

**Questions:**

- for a fixed commutating parametrisation $G$, is it possible to understand the effect of $x_{init}$ on the associated mirror map $R$
- you provide sufficent conditions, do you think they are necessary ? or not ?
- is Nash's embedding constructive ? can we know F, G, or the dimension D ?
- do you think the claim in Remark 4.7 still holds if we restrict ourselves to losses which are non time-dependent ?



**Strengths And Weaknesses:**

Overall I think this is a very good paper.  It is very well written:
- the motivations are clear and relevant as it sheds light on when we can hope to obtain implicit regularisation results using mirror flow type of analysis
- the setup and preliminaries are very clear
- All the results are introduced and commented, examples are also provided
- though I am not familiar with all the mentioned notions, they all seem well defined and the math seems rigorous

The paper is very well written and I believe the general framework they introduce as well as the lemmas / tools they provide are relevant: as they help understand the link between gradient flow, reparametrisation and Mirror flow.


Minor comments:
- in the main contributions in the intro: "we also five new convergence analysis in such settings, filling the gap in previous works [22, 57, 8] where parameter convergence is only assumed but not proved." is an overclaim as the convergence of mirror descent / flow is well understood and has been investigated for years (though I agree that the mentioned works seem to have overlooked them.
- I would have appreciated an example illustrating Lemma 4.5 and Lemma 4.6 , for the $u^2 - v^2$ parametrisation (or even simply $G(x) = x$)  for example: what is $\Psi(x, \mu)$ in this case ? and $\mu_t$? and $G(\Psi(x_{init}, \mu))$ ? this would help the reader gain more intuition / understanding

---

> ### Author Response · Authors · 2022-08-02
> **Response to Reviewer eah9**
>
> We would like to thank the reviewer for the positive support and also for the constructive feedback. Below we provide responses to the comments and questions.
>
> **Q1**: The statement about the contribution in terms of the convergence analysis is an overclaim as the convergence of mirror descent/flow is well understood and has been investigated for years.
>
> **A1**: We are not claiming convergence of mirror-descent per se is new. By establishing the equivalence between reparametrized gradient flow and mirror flow, we can leverage Alvarez et al. (2004) to show convergence of trajectory of gradient flow for objectives that are non-convex.
>
> ---
>
> **Q2**: An example illustrating Lemma 4.5 and Lemma 4.6. For the $u^{\odot 2}−v^{\odot 2}$ parametrisation (or even simply $G(x)=x$) for example: what are $\psi(x,\mu)$, $\mu_t$, $G(\psi(x_{init},\mu))$ in this case?
>
> **A2**: Let’s consider the $u^{\odot 2}−v^{\odot 2}$ parametrization. As in the proof of Corollary 4.12, it is straightforward to check that in this case $\psi(x_{init}, \mu) = \binom{u_0 \odot e^{-2\mu}}{v_0 \odot e^{2\mu}}$ and $G(\psi(x_{init}, \mu)) = u_0^{\odot 2} \odot e^{-4\mu} - v_0^{\odot 2} \odot e^{4\mu}$. $\mu_t$ depends on the loss, and there is no closed-form formula in general.
>
> ---
>
> **Q3**: For a fixed commutating parametrisation $G$, is it possible to understand the effect of $x_{init}$ on the associated mirror map $R$?
>
> **A3**: This is a great question! For the concrete parametrization $G(x) = u^{\odot 2} - v^{\odot 2}$, the effect of initialization is studied in Woodworth et al. (2020). It’s still open if we can study the effect of initialization in an abstract way, i.e., without knowing the exact expression of $G$.
>
> _Reference_:
>
> - Blake Woodworth, Suriya Gunasekar, Jason D Lee, Edward Moroshko, Pedro Savarese, Itay Golan, Daniel Soudry, and Nathan Srebro. Kernel and rich regimes in overparametrized models. In Conference on Learning Theory, pages 3635–3673. PMLR, 2020
>
> ---
>
> **Q4**: You provide sufficient conditions, do you think they are necessary? or not?
>
> **A4**: We provide necessary conditions in the rebuttal revision and conjecture the necessary conditions are indeed sufficient. Specifically, the necessary condition is stated as follows: $\mathrm{Lie}^{\geq 2}(\partial G) \big\vert_x \subseteq \mathrm{ker}(\partial G(x)), \forall x \in M$ where $\mathrm{Lie}^{\geq K}(\partial G) := \mathrm{span} \{[[[[\nabla G_{j_1}, \nabla G_{j_2}], \ldots], \nabla G_{j_{k-1}}], \nabla G_{j_k}] \mid k\geq K, \forall i \in [k], j_i \in [d] \}$. This is condition is quite abstract, but in essence, it requires that moving along the direction of any Lie bracket generated by $\{\nabla G_i\}_{i=1}^d$ does not change the value of $\partial G(x)$.
>
> ---
>
> **Q5**: Is Nash's embedding constructive? Can we know $F$, $G$, or the dimension $D$?
>
> **A5**: Nash’s Embedding Theorem is indeed a constructive result, and it only requires $D \leq \max(\frac{d(d+5)}{2}, \frac{d(d+3)}{2} + 5)$. (Though it doesn’t admit an analytic formula.)
>
> ---
>
> **Q6**: Do you think the claim in Remark 4.7 still holds if we restrict ourselves to losses which are non time-dependent?
>
> **A6**: This is an excellent question! We conjecture that it is true but don’t have a proof yet.

---

### Official Review · Reviewer_94Rv · 2022-07-15

**Rating:** 5
**Confidence:** 4
**Soundness:** 3 good
**Presentation:** 3 good
**Contribution:** 2 fair

**Summary:**

This work establishes a link between minimizing a loss function using continuous-time mirror descent (i.e. mirror flow) and minimizing a reparameterized version of the same loss function using continuous-time gradient descent (i.e. gradient flow) to explain the implicit bias of gradient flow in reparameterized models. The authors introduce a class of reparameterization mappings named commuting parameterizations which satisfy certain commutativity conditions of the Jacobian. The authors show that for any commuting parameterization, there exists a mirror function in the original domain such that the trajectory of gradient flow of a loss function with that parameterization coincides with the trajectory of mirror flow of the same loss function in the original domain with the associated mirror function. Furthermore,  they also give a converse result by showing the existence of a commuting parameterization for any mirror function such that gradient flow in the reparameterized space is equivalent to mirror flow in the original space. The authors also show that several past works on implicit bias/regularization can be considered as specific cases of different commuting parameterizations.


**Questions:**

I would appreciate it if the authors clarify the following points:

1. Is it possible to provide an order-wise relationship between $D$ and $d$? More explicitly, given mirror function $R$, can you tell how large the dimension of the reparameterization space $D$ must be with respect to $d$?
2. Given a mirror function $R$, how easy and feasible is it to construct a commuting parameterization $G$?
3. Does it make a difference in terms of finite-time (i.e. early stopping) convergence rate if one performs the training in x-space with gradient flow or w-space with mirror flow?


**Limitations:**

The authors could make this paper more impactful if they can provide a discrete-time analysis or a finite-time (i.e. early stopping) convergence rate analysis. It would also be interesting to know the order-wise relationship between dimensions $d$ and $D$. The authors could also support their results by simulations performed with practically relevant models (deep nets, kernel models, etc.) to show efficiency of using (possibly sparse) parametrizations in optimizing these models.

**Strengths And Weaknesses:**

Strengths:
1. The authors give a sufficient condition to check whether gradient flow for a given reparametrization can be written as a mirror flow for some mirror map in a possibly smaller dimensional domain. The condition simply relies on checking commutativity of the row vectors of the Jacobian of reparametrization map $G:M\subset \mathbb{R}^{D} \to \mathbb {R}^d$, i.e., $[\nabla G_i, \nabla G_j] (x)=0$ for any $i,j\in[d]$ for all $x \in M$. They also show that several examples of reparameterization satisfy this condition.

Weaknesses:
1. The nature of the proof of Theorem 4.4 seems to be existential rather than constructive. It doesn't seem to be straightforward to construct a commuting reparametrization $G$ given a mirror map $R$ and vice versa.
2. Although the analysis is valid for continuous-time flows, the authors do not address the fidelity of this equivalence in discrete-time setting which is more practically relevant.
3. The theoretical analysis in this paper could have been supported with extensive simulations on certain benchmarks.


Although the paper is well written and easy to understand, it mostly builds on Amid and Warmuth (2020) by generalizing to non-injective parametrizations. This makes it an incremental contribution as there is neither much technical novelty nor extensive simulations to show practical relevance and to draw more insights.

---

> ### Author Response · Authors · 2022-08-02
> **Response to Reviewer 94Rv**
>
> We would like to thank the reviewer for the careful reading and thoughtful feedback. Please see our responses to the questions and comments below.
>
> **Q1**: The proof of Theorem 4.4 is existential rather than constructive.
>
> **A1**: Actually Theorem 4.4 is constructive and the mirror map is given by Lemma 4.6. We will highlight this in the revision.
>
> ---
>
> **Q2**: It doesn’t seem to be straightforward to construct a commuting reparametrization $G$ given a mirror map $R$ and vice versa. Given a mirror function $R$, how easy and feasible is it to construct a commuting parametrization G?
>
> **A2**: Indeed the embedding constructed by Nash’s Embedding Theorem is not explicit and in general infeasible to compute. It is still open if there is a simpler embedding when the metric is Hessian of some convex function. But for standard convex functions $R$ which are typically separable, like negative entropy and Burg’s entropy, the expression for G is straightforward to compute. (See discussion in Appendix E.1 in the rebuttal revision.)
>
> ---
>
> **Q3**: The authors do not address the fidelity of this equivalence in discrete-time setting which is more practically relevant.
>
> **A3**: For GD with small learning rate $\eta$, a standard ODE approximation result implies that the distance between discrete and continuous dynamics is $O(\eta)$. We suggest the reviewer refer to the recent preprint [Ghai et al, 2022] for optimization analysis of discretized dynamics, which shows the discrete and continuous dynamics are close.
>
> _Reference_:
>
> - Ghai, Udaya, Zhou Lu, and Elad Hazan. "Non-convex online learning via algorithmic equivalence." arXiv preprint arXiv:2205.15235 (2022).
>
> ---
>
> **Q4**: The paper mostly builds on Amid & Warmuth (2020) by generalizing to non-injective parametrizations. This makes an incremental contribution as there is neither much technical novelty nor extensive simulations to show practical relevance and to draw more insights.
>
> **A4**: We respectfully disagree. Generalization to non-injective parametrizations is significant contributions because injective parametrization which preserves dimension is very restrictive, for both directions of the equivalence between reparametrized gradient flow (GF) and mirror flow (MF).
>
> 1. **GF -> MF**: Amid & Warmuth (2020) make a very strong assumption that all preimages $x$ of $w$ under parametrization G have the same $\partial G(x)\partial G(x)^\top$ and this is equal to Hessian of a convex function. This assumption excludes even the simplest and most canonical example, $w= u^{\odot 2} - v^{\odot 2}$ of Woodworth et al. (2020). Thus their assumption is clearly not capturing the phenomenon being studied. Our results showed that such a strong assumption is not necessary by leveraging tools from differential geometry. Furthermore, our framework suggests a new assumption (which does not resemble past assumptions, we think, and is very easy to check when it holds).
>
> 2. **MF -> GF**: We also show that our new assumption, namely commuting parametrization, exactly characterizes the phenomenon being studied by exhibiting in the sense that every mirror descent can be obtained by an overparametrization that satisfies our assumptions. However, given an arbitrary convex function $R$, it’s in general **impossible** to find an **injective** parametrization $G: \mathbb{R}^d\to\mathbb{R}^d$ such that gradient flow with respect to $G$ is equivalent to mirror flow with respect to $R$. This is because the existence of such G implies that the Riemannian manifold $(\mathbb{R}^d,\nabla ^2 R)$ is flat, that is, it can be isometrically embedded into $(\mathbb{R}^d, I_d)$, which is in general false. For example, let $R:\mathbb{R}^3\to \mathbb{R}$ be defined as $R(w) = |w|_2^4$, and we can compute the scalar curvature of $(\mathbb{R}^3,\nabla ^2 R)$ at $(1,1,1)$ is $0.0741$, meaning the Riemannian manifold is not flat because flat manifolds have $0$ scalar curvature everywhere.
>
> ---
>
> **Q5**: Is it possible to provide an order-wise relationship between $d$ and $D$?
>
> **A5**: Yes, according to Nash’s Embedding Theorem, it only requires $D \leq \max(\frac{d(d+5)}{2}, \frac{d(d+3)}{2} + 5)$. In the revision, we will explicitly write out this relationship between $d$ and $D$.
>
> ---
>
> **Q6**: Does it make a difference in terms of finite-time (i.e. early stopping) convergence rate if one performs the training in x-space with gradient flow or w-space with mirror flow?
>
> **A6**: No, since the equivalence established in Theorem 4.4 between mirror flow and reparametrized gradient flow is throughout the entire trajectory, i.e., $w(t) = G(x(t))$ for all $t \geq 0$.

---

> > ### Comment · Reviewer_94Rv · 2022-08-08
> > **Re: Response to Reviewer 94Rv**
> >
> > Dear authors,
> >
> > Thank you very much for taking the time and putting in the effort to answer my questions. I find the explanation regarding the constructive nature of Nash's Embedding Theorem and the relationship between $D$ and $d$ (Qs 1,2,5) as well as the recent result addressing discrete-time approximation of OMD as OGD (Q3).
> >
> > I appreciate the author's answer to Q4 and their view on the significance of the noninjective setting. Although I agree with the points they raised and appreciate the effort they put into this work, I still find it hard to convince myself to claim that this is a very significant contribution compared to prior works.
> >
> > Overall, I am willing to increase my rating to a borderline accept as the authors clarified certain points in their answers and in their final version. As a final note, I would suggest the authors discuss the necessary condition (theorem E.3) and the restrictiveness of the injective setting in the main text.

---

> > > ### Author Response · Authors · 2022-08-09
> > > **Thanks for the response!**
> > >
> > > We thank the reviewer for the positive feedback! As suggested by the reviewer, we will add the discussion on the necessary condition and the restrictiveness of the injective setting to the main text in the future version.
> > >
> > > Also, we kindly remind the reviewer that the rating seems still unchanged in the system.

---

> ### Author Response · Authors · 2022-08-08
> **Follow up with Reviewer 94Rv**
>
> Dear Reviewer 94Rv,
>
> Since the author-reviewer discussion period is ending soon, we would like to follow up with you to see if you have any further questions.
> We believe we have addressed your main questions in our rebuttal. We are looking forward to your feedback.
>
> Thank you,
>
> Authors

---

### Author Response · Authors · 2022-08-02
**Summary of rebuttal revision**

First, we would like to thank all the reviewers for their careful reading and insightful comments and questions.

To further complement our response to the reviewers, we make the following major updates in the rebuttal revision:

1. We revise the statement of Theorem 4.4 and Theorem 4.11 by emphasizing that the mirror map $R$ is explicitly given by Lemma 4.6. This is in response to Reviewer 94Rv’s comment on Theorem 4.4 and Reviewer puj1’s question on whether the convex regularizer $R$ in Theorem 4.11 is implicit or not.
2. We revise the statement of Theorem 5.2 by clarifying the quantitative relationship between $d$ and $D$. This addresses Reviewer 94Rv’s question on the order-wise relationship between $d$ and $D$ and Reviewer eah9’s question on Nash’s embedding theorem.
3. We add Appendix E.1 which discusses how to construct the parametrization $G$ from a given mirror map $R$. This is in response to Reviewer 94Rv’s related questions and is also related to Reviewer eah9’s question on Nash’s embedding theorem.
4. We add analysis and discussion on the example of matrix factorization in Appendix E.2 and E.3. This corresponds to Reviewer puj1’s question on how restrictive is the assumption of commuting parametrizations.
5. We provide a necessary condition on smooth parametrization to be commuting, Theorem E.3 in Appendix E.3, and we conjecture that it is indeed sufficient in Conjecture E.4. The proof of Theorem E.3 is provided in Appendix E.4. This addresses Reviewer eah9’s question about the necessary conditions, and is also related to Reviewer puj1’s question on the restrictiveness of the model at study.

All these changes are marked red in the PDF.

Thank you,

Authors

---

### Meta-Review · Area_Chair_BhpH · 2022-09-03

**Recommendation:** Accept
**Confidence:** Certain

**Metareview:**

The paper characterizes how reparametrized/overparametrized models can be understood as a version of mirror descent on a different objective through a notion they call commuting parameterization. The reviews are all positive and agree that the paper improves our understanding of this phenomenon.

**Award:**

No

---

### Decision · Program_Chairs · 2022-09-14

Accept